

# Implementing a dynamic representation of fire and harvest including subgrid-scale heterogeneity in the tile-based land surface model CLASSIC v1.45

Salvatore R. Curasi[1,2,https://orcid.org/0000-0002-4534-3344], Joe R. Melton[1,https://orcid.org/0000-0002-9414-064X], Elyn R. Humphreys[2,https://orcid.org/0000-0002-5397-2802], Txomin Hermosilla[3,https://orcid.org/0000-0002-5445-0360], Michael A. Wulder[3,https://orcid.org/0000-0002-6942-1896]

[1]Climate Research Division, Environment, and Climate Change Canada, Victoria, BC, V8N 1V8, Canada
[2]Department of Geography & Environmental Studies, Carleton University, Ottawa, ON, K1S 5B6, Canada
[3]Canadian Forest Service (Pacific Forestry Centre), Natural Resources Canada, Victoria, BC, V8Z 1M5, Canada

*Correspondence to*: Salvatore R. Curasi (sal.curasi@ec.gc.ca)

**Abstract.** Canada's forests play a critical role in the global carbon (C) cycle and are responding to unprecedented climate change as well as ongoing natural and anthropogenic disturbances. However, the representation of disturbance in boreal regions is limited in pre-existing land surface models (LSMs). Moreover, many LSMs do not explicitly represent subgrid-scale heterogeneity resulting from disturbance. To address these limitations, we implement harvest and wildfire forcings in the Canadian Land Surface Scheme Including Biogeochemical Cycles (CLASSIC) land surface model alongside dynamic tiling that represents subgrid-scale heterogeneity due to disturbance. The disturbances are captured using 30-m spatial resolution satellite data (Landsat) on an annual basis for 33 years. Using the pan-Canadian domain (i.e. all of Canada south of 76°N) as our study area for demonstration, we determine the model setup that optimally balances detailed process representation and computational efficiency. We then demonstrate the impacts of subgrid-scale heterogeneity relative to standard average individual-based representations of disturbance and explore the resultant model biases. Our results indicate that the modeling approach implemented can balance model complexity and computational cost to represent the impacts of subgrid-scale heterogeneity resulting from disturbance. Subgrid-scale heterogeneity is shown to have impacts 1.5 to 4 times the impact of disturbance alone on gross primary productivity, autotrophic respiration, and surface energy balance processes in our simulations. These impacts are a result of subgrid-scale heterogeneity slowing vegetation re-growth and affecting surface energy balance in recently disturbed, sparsely vegetated, and often snow-covered fractions of the land surface. Representing subgrid-scale heterogeneity is key to more accurately representing timber harvest, which preferentially impacts larger trees on higher quality and more accessible sites. Our results show how different discretization schemes can impact model biases resulting from the representation of disturbance. These insights, along with our implementation of dynamic tiling may apply to other tile-based LSMs. Ultimately our results enhance our understanding of, and ability to, represent disturbance within Canada to facilitate a comprehensive process-based assessment of Canada's terrestrial C cycle.



**Copyright statement**

## 1 Introduction

Canada's forests play a critical role in the global carbon (C) cycle (Keenan and Williams, 2018; Lenton et al., 2008). Canada's forests are also responding to both unprecedented climate change and on-going anthropogenic disturbance (Lenton et al., 2008; White et al., 2017). Unfortunately, disentangling the relative impacts of disturbance processes and climate change on the Canadian forest C cycle is difficult (Sulla-Menashe et al., 2018; Goetz et al., 2005; Ju and Masek, 2016; Weber and Flannigan, 1997). Process-based land surface models (LSMs) provide a tool to evaluate the impacts of both types of disturbance but there has only been limited representation of anthropogenic disturbance in regional or global C cycling assessments (Friedlingstein et al., 2019; Peng et al., 2014; Chaste et al., 2017; Hayes et al., 2012). Moreover, of those LSMs that do explicitly represent anthropogenic disturbance, only a small subset account for the resulting subgrid-scale heterogeneity (Le Quéré et al., 2018; Nabel et al., 2020; Pongratz et al., 2018). Here we demonstrate the impacts of disturbance and sub-grid scale heterogeneity on C and energy fluxes, by implementing a dynamic tiling scheme in the Canadian Land Surface Scheme Including Biogeochemical Cycles (CLASSIC).

Subgrid-scale landscape heterogeneity refers to any characteristic of the landscape that differs at scales below that of the main model grid, in this case, differences in tree age and biomass in burned or harvested subfraction of the grid cell. Tile-based LSMs, unlike individual-based models (i.e. models which simulate the landscape using several heterogeneous individuals), do not inherently represent subgrid-scale heterogeneity. Instead, the tile represents the average individual of a given plant function type (PFT) and thus represents the PFT's average state within the grid cell (e.g. a single height, biomass, etc.), which is used to simulate fluxes. Although most tile-based LSMs account for wood harvest, few represent the resulting subgrid-scale landscape heterogeneity and rather represent disturbances impact on the average individual PFT (Le Quéré et al., 2018; Pongratz et al., 2018; Nabel et al., 2020).

Stand-replacing forest disturbances (i.e. timber harvest and fire) directly impact forest C stocks through the removal of standing biomass (Wulder et al., 2020). In addition, stand-replacing disturbances also impact stand structure, especially in the case of managed timber harvest (Pan et al., 2010; Kuuluvainen and Gauthier, 2018; Pan et al., 2013). The resulting stand structure impacts forest function such as the the exchange of matter and energy with the atmosphere as well as forest





response to climate change (Erb et al., 2017; Luyssaert et al., 2014; Körner, 2006; Dore et al., 2010; Liu, 2005; Maness et al., 2012; Hirano et al., 2017). Historically 0.4% of Canada's ~650 Mh of forested ecosystems are affected by stand-replacing disturbance per year (White et al., 2017). The age structure of Canadian forests due to historical disturbance has impacted the
70 strength of the historical C sink in Canadian forests (Kurz and Apps, 1999, 1993; Böttcher et al., 2008). Age structure resulting from disturbance also influences the surface energy balance of stands for example by altering sensible heat flux due to differences in snow cover and albedo and altering the seasonality of surface energy budgets and land surface properties (Liu, 2005; Maness et al., 2012). Therefore, it is key that we enhance our ability to accurately represent both disturbance processes and the influence of subgrid-scale heterogeneity that disturbances produce within LSMs.

Disturbance events impact the response of Canada's forests to climate change. The response of forest productivity, forest soil decomposition processes, and evaporation rates to warming, rising $CO_2$ concentrations, and changes to precipitation regimes will depend on stand structural characteristics and tree species characteristics (Hember et al., 2012; Kurz et al., 1997; Körner, 2006; Shrestha and Chen, 2010; Bond-Lamberty and Gower, 2008; Czimczik et al., 2006; Kurz et al., 2008).
Warmer temperatures and higher atmospheric $CO_2$ concentrations are likely to increase the productivity of boreal forests, whereas drought stress and changing disturbance regimes are likely to decrease productivity and enhance the decomposition of soil C leading to a patchwork of contrasting future responses (Babst et al., 2019; Reich et al., 2018; Lenton et al., 2008; Weber and Flannigan, 1997; Potapov et al., 2008; Ju and Chen, 2008; Sulla-Menashe et al., 2018). Complex changes in vegetation productivity have already been observed across the pan-Canadian domain due to the intermingling of different
disturbance regimes and different vegetation sensitivity to climate change (Marchand et al., 2018; D'Orangeville et al., 2018; Ma et al., 2012; Girardin et al., 2016). Decreases in vegetation productivity are generally occurring in northwestern boreal forests, whereas southeastern boreal forests show positive trends (Marchand et al., 2018). Much of the landscape scale change in vegetation productivity detected across Canada's boreal forests is a product of, or influenced by, stand-replacing disturbance (Hermosilla et al., 2015b, a). Some negative productivity trends in the southern fringes of western undisturbed
forests can largely be attributed to moisture stress and some of the positive trends in cooler and wetter portions of eastern boreal forests can be attributed to warming (Marchand et al., 2018; Sulla-Menashe et al., 2018). Process-based models which represent both disturbance and the resultant subgrid-scale landscape heterogeneity can offer insight into the drivers of these complex trends (Böttcher et al., 2008).

CLASSIC is a tile-based LSM that can be coupled to the Canadian Earth System Model (CanESM). Several methods are available for representing disturbance history in tile-based LSMs. Some models represent the age classes within the stand using a fixed number of tiles to represent fractional areas below the scale of the main model grid (i.e. from 2 to 12 tiles) (Shevliakova et al., 2009; Yue et al., 2018a; Naudts et al., 2015; Yang et al., 2010; Stocker et al., 2014). Alternatively, several models simulate subgrid scale forest structure using another model housed in a separate module coupled to the main
model (Bellassen et al., 2010; Haverd et al., 2014). The module takes information about net primary productivity from the





main model and uses it to simulate and track the growth of individual trees. The module then returns grid cell average state information (i.e. biomass and litter fluxes) which is used by the main model to simulate subsequent fluxes. Finally, a recently developed approach uses a fixed number of tiles to represent age classes (Nabel et al., 2020). Tile fractional area and associated state variables (i.e. biomass C) are horizontally exchanged between the tiles to represent processes like aging, 105 harvest, and disturbance. Each approach entails a host of strengths and weaknesses as well as its own biases resulting from discretization error (Nabel et al., 2020; Fisher et al., 2018).

In this study, to demonstrate the impacts of disturbance and sub-grid scale heterogeneity on C and energy fluxes, we implement a dynamic tiling scheme in CLASSIC. Our implementation is a modified version of approaches that use a fixed 110 number of tiles to represent age classes within the stand and may apply to other tile-based LSMs. We build upon a version of CLASSIC tailored to the pan-Canadian domain using region-specific plant functional types (PFTs) and a 0.22º (~20 km x 20 km) common grid (Curasi et al., 2022). The age classes within the stand are represented using a variable number of sub-grid tiles of variable fractional area and subject to a user-determined maximum number of tiles available for the simulation. Tiles are split to represent disturbance and the resulting age and size structures. Tiles, and their underlying characteristics, are 115 joined by the simulation either when the number of tiles reaches the user-determined maximum bound or preemptively based upon other user-determined parameters. The model is driven by externally forced harvest and fire from region-specific disturbance history drivers. We set an optimal maximum number of tiles available for the simulation by evaluating different model setups through model-on-model evaluation and assessing the run time of these setups. Finally, we compare the biases across runs to assess the impacts of the imposed trade-off between run time and a more detailed representation (i.e. more 120 tiles). This investigation provides insight into the setup and role of these processes within CLASSIC, as a step towards a comprehensive process-based assessment of Canada's terrestrial C cycle. These insights may also apply to other tile base LSMs.

## 2 Methods

### 2.1 Study area

We use a domain, which encompasses all of Canada south of 76°N as our study area for demonstration. Canada contains 650 Mha of forested land and 98 Mha (18%) of this forested land was disturbed from 1985 - 2015. On average each year 1.61 Mha is disturbed by wildfire, whereas 0.64 Mha is disturbed by harvest (Hermosilla et al., 2019). Disturbance due to wildfire is most prevalent in northern boreal regions, whereas harvest and other anthropogenic disturbances are more common in southern boreal regions where wildfire is suppressed. The spatial extent of individual disturbances is highly variable. In 130 Canada Timber harvest change objects (i.e. a contiguous harvest event occurring within a single year) involve the removal of 98 ± 115 ha of trees on average. These timber harvest patterns are heavily influenced by forest management practices (Hermosilla et al., 2015b; White et al., 2017). Similarly, fire change objects (i.e. a contiguous fire event occurring within a



single year) burn 324 ± 633 ha on average (Hermosilla et al., 2015b). The spatial scale of these change objects, sourced from 30-m spatial resolution Landsat imagery, falls well below the ~40,000 ha resolution of the 0.22° pan-Canadian domain

model grid. Located largely in southern latitudes, around 52% of Canada's forested land is considered managed forest (Stinson et al., 2011). Canada's forest structure is characterized by relatively young stands in central and northwestern Canada, with much older stands found in the Pacific coastal and interior forests in British Columbia (Maltman et al., 2023). Forest ages in Canada are the result of prevailing natural disturbance regimes and, to a lesser extent, forest management practices (Pan et al., 2013).

**2.2 The CLASSIC model**

CLASSIC is an open-source community model that builds upon the coupled Canadian Land Surface Scheme (CLASS) (Verseghy, 2000, 2017; Verseghy et al., 1993; Verseghy, 2007) and the Canadian Terrestrial Ecosystem Model (CTEM) (Melton and Arora, 2016; Arora, 2003). CLASSIC v1.0 is described and evaluated by Melton et al., (2020) and Seiler et al., (2021). A detailed description of model updates and improvements to CLASSIC since v1.0 that are utilized by our

simulations can be found in Asaadi et al. (2018), MacKay et al. (2022), and Curasi et al. (2022). We carry out simulations of the pan-Canadian domain using a parameterization of the model which includes Canada-specific plant functional types (PFTs) that were developed and evaluated by Curasi et al., (2022).

The CTEM dynamic vegetation sub-model simulates photosynthetic fluxes, at a thirty-minute time step in offline

simulations, and the allocation of C within live vegetation to structural and non-structural components of leaves, stems, and roots at a daily time step. CTEM also simulates daily autotrophic respiration from leaves, stems, and roots and heterotrophic respiration fluxes from litter and soil C. The pan-Canadian parameterization of CTEM utilizes fourteen biogeochemical PFTs (Curasi et al., 2022). CTEM is coupled to CLASS on a daily time step and provides CLASS with vegetation height, leaf area index, biomass, and rooting depth. CLASS, in turn, provides CTEM with mean daily soil moisture, soil

temperature, and net radiation incident on the land surface. CLASS simulates energy exchange from four possible subareas: bare ground, snow-covered bare ground, canopy-covered ground, and snow-covered canopy, on a thirty-minute time step. It uses 20 ground layers from 0.1 m to 30 m thick to a depth of over 61 m and simulates heat transfer within all permeable soil layers and the underlying bedrock. It also simulates water fluxes between the soil layers up until the depth of the impermeable bedrock layer, derived from Shangguan et al. (2017). CLASS models a single-layer canopy and uses five

physics PFTs, which map directly onto the 14 CTEM biogeochemical PFTs, in the pan-Canadian parameterization (Curasi et al., 2022).

**2.3 Dynamic tile representation of externally forced fire and harvest**

CLASSIC can utilize either a composite (1 tile) or mosaic (>1 tile) representation of the land surface. The composite representation simulates average individual PFTs for each grid cell and uses their average structural attributes (i.e. leaf area





index, height, and rooting depth) to simulate the energy balance, and physical environment (i.e. soil temperature). The structural attributes of all of the average individual PFTs that exist within a grid cell are averaged in proportion to their fractional coverages and the PFTs all experience a common land surface physical environment. For the composite representation, a disturbance event (i.e. wood harvest) takes C from the average individual PFTs pools proportional to the areal fraction disturbed (i.e. a complete harvest of 50% of the grid cell thereby removing 50% of the vegetation biomass;

Figure 1).

The mosaic representation splits the grid cell into multiple tiles representing fractional areas of the grid cell. Each tile receives the same meteorological forcing but simulates its respective average individual of each PFT present, structural attributes, and energy balance. The structural attributes of all of the average individual PFTs that exist within each tile are

averaged in proportion to their fractional coverages and the PFTs all experience a land surface physical environment common to that tile. The tiles are aggregated to the scale of the final model grid by accounting for each tile's fractional coverage of the grid cell. This tiling capability has been used in the past to investigate the impacts of subgrid-scale heterogeneity in soil texture (Melton et al., 2017), and vegetation cover (Melton and Arora, 2014; Li and Arora, 2012), as well as impacts on competition (Shrestha et al., 2016). We adapt the mosaic representation to dynamically create disturbance

history tiles and represent the subgrid-scale heterogeneity resulting from disturbance (i.e. represent a complete harvest of an area corresponding to 50% of the grid cell as a 100% reduction of the vegetation biomass in a newly created subgrid tile that covers 50% of the grid cell; Figure 1).

### 2.3.1 Notation and background

We present generalized equations that illustrate the dynamic tiling calculations done by the model to split and join tiles. In

these equations, scalars are lowercase letters (i.e. x = [1]), vectors are bold lowercase letters (i.e. $\mathbf{x}$ = [x1, x2, …, xn]), and

matrices are bold uppercase letters (i.e. $\mathbf{X} = \begin{pmatrix} x_{1,1} & \cdots & x_{1,n} \\ \vdots & \ddots & \vdots \\ x_{n,1} & \cdots & x_{n,n} \end{pmatrix}$). The model is set up to simulate state variables for a user-

defined maximum number of tiles within a grid cell (i.e. the state variable $\mathbf{x_{all}}$ with a length equal to the user-determined maximum number of tiles). Tiles can be set as either active or inactive in a given timestep (Figure 1). When the model identifies active tiles for merging or splitting (e.g. sections *2.3.2, 2.3.4,* and *2.3.5*) they become candidate tiles. Depending

upon the operation and the fraction of the grid cell involved, anywhere between one and the total number of tiles being actively simulated are candidate tiles for the merging or splitting operation. Because the maximum number of tiles within a grid cell is defined at the start of the run the model's integrated software routines ensure that the number of tiles being actively simulated is always less than or equal to the maximum number of tiles the model is set up to simulate minus the number of tiles needed to simulate disturbance events in that simulation year (i.e. up to 1 fire event and 1 harvest event per

year; see section 2.3.3). During the merging or splitting operation the model temporarily stores values from the candidate





tiles before the operation (i.e. $\mathbf{x_{pre}}$ of length n candidate tiles) and after the operation (i.e. $\mathbf{x_{post}}$ of length n candidate tiles) and uses them to calculate the values for a new single output tile (i.e. $x_{new}$). All of these values are temporarily stored and used in calculations up until the point where the dynamic tiling operation is complete, and the model's main data structures are updated.


### 2.3.2 Dynamic tiling splits and joins

Dynamic tiling allows the model to split grid cells into subgrid tiles during the model run, or join existing subgrid tiles. Dynamic tiling operations (splitting/joining) occur on January 1st at the annual time step alongside rigorous checks to ensure water, mass, and energy conservation. The area occupied by a given tile is a fraction of the grid cell land area between zero

and one (i.e. $\mathbf{a_{all}}$ with length equal to the user-determined maximum number of tiles). The sum of $\mathbf{a_{all}}$ for all the active tiles within a grid cell must equal one. When tiles are split the fractional area occupied by the single new tile ($a_{new}$) cannot exceed the sum of the vector of fractional areas of the candidate tiles ($\mathbf{a_{pre}}$ of length $n$ candidate tiles). The candidate tiles' fractional areas are a product of the dynamic tiling operations that occur in all previous time steps. When the first dynamic tiling operation in a run occurs $a_{pre} = [1]$, but $\mathbf{a_{pre}}$ is a much more complex vector in subsequent operations (e.g. $\mathbf{a_{pre}} = [0.1, 0.2,$

$0.3]$). The candidate tiles are later assigned a vector of new fractional areas adjusted to account for the creation of the new tile and the decrease in size of the candidate tiles ($\mathbf{a_{post}}$ also of length $n$ candidate tiles; Eqn. 1).

$$a_{new} \leq \sum_{i=1}^{n} a_{pre,i}$$

$$a_{post,n} = a_{pre,n} - a_{new} \frac{a_{pre,n}}{\sum_{i=1}^{n} a_{pre,i}} \tag{1}$$

When tiles are joined by the model the fractional area of the new tile is the sum of the vector of the fractional areas of the

candidate tiles. The candidate tiles are later assigned fractional areas of zero (Eqn. 2).

$$a_{new} = \sum_{i=1}^{n} a_{pre,i}$$

$$\boldsymbol{a_{post}} = 0 \tag{2}$$

For a tile or group of tiles to be split or joined they must pass rigorous checks that ensure they share the same abiotic characteristics and prescribed fractional PFT cover. These characteristics (i.e. soil texture, soil permeable depth, and PFT

fractional coverage) are copied directly to the new tile by the split or join. Mass-based variables (i.e. vegetation C pool mass, soil C pool mass, soil water, ponded water, and water held in the vegetation canopy) are split or joined using fractional area-based weighted averages to ensure mass balance. The value of the mass-based variable in the new tile ($\mathbf{M_{new}}$ for $l$ layers and $o$ PFTs; kg m$^{-2}$) is the average of the values for the candidate tiles ($\mathbf{M_{pre}}$ of length $n$ candidate tiles for $l$ layers and o PFTs; kg m$^{-2}$) weighted by the fractional areas of the candidate tiles (Eqn. 3).





$$m_{new,lo} = \frac{\sum_{i=1}^{n} m_{pre,ilo}\, a_{pre,i}}{\sum_{i=1}^{n} a_{pre,i}} \tag{3}$$

Temperature-based variables (i.e. temperatures of the vegetation canopy, ponded water, snowpack, and soil) are split or joined using a fractional area-based weighted average that blends the different temperature materials from the candidate tiles. The value of a given temperature for the new tile ($\mathbf{t_{new}}$ for $l$ layers; K) is a function of the temperatures in the candidate tiles ($\mathbf{T_{pre}}$ of length $n$ candidate tiles for $l$ layers; K) weighted by the fractional areas of the candidate tiles before the split, the masses of the pools which track that temperature ($\mathbf{M_{pre}}$ of length $n$ candidate tiles for $l$ layers and $m$ pools; kg m$^{-2}$), and the specific heat capacities which characterize those mass pools ($\mathbf{c}$ for $m$ pools; J kg$^{-1}$ K$^{-1}$; Eqn. 4).

$$t_{new,l} = \frac{\sum_{i=1}^{n}\left(t_{pre,il}\, a_{pre,i} \sum_{j=1}^{m}(m_{pre,ilj}\, c_j)\right)}{\sum_{i=1}^{n}\left(t_{pre,il}\, a_{pre,i} \sum_{j=1}^{m}(m_{pre,ilj}\, c_j)\right)} \tag{4}$$

### 2.3.3 Dynamic tiling management

The maximum number of dynamic tiles in a given simulation is limited by parameters set at the start of the model run. If this upper limit is reached, tiles are joined based on those tiles that pass the criteria for acceptable similarity. By default, the model joins the most similar tiles within the grid cell based on the tiles' vegetation heights. The model uses the vector of tile average vegetation heights ($\hbar$ of length $n$ total number of tiles; m) to calculate the absolute difference matrix of tile average vegetation heights ($\mathbf{\Delta H}$ an $n$ total number of tiles by $n$ total number of tiles matrix; m) to judge the similarity between tiles. The tile average vegetation height is a function of each PFT height ($\mathbf{H}$ of length $n$ total number of tiles for $o$ PFTs; m) and the PFT fractional coverage within the tile ($\mathbf{F}$ of length $n$ total number of tiles for $o$ PFTs; Eqn 5; Figure S1).

$$\Delta\hbar_{n1,n2} = |\hbar_{n1} - \hbar_{n2}|$$

$$\hbar_n = \frac{\sum_{k=1}^{o} h_{n,k}\, f_{n,k}}{\sum_{k=1}^{o} f_{n,k}} \tag{5}$$

Two optional model parameters allow for tiles to be preemptively joined before reaching the maximum number of dynamic tiles. First, when set the relative height threshold ($rht$; unitless) is used to calculate a threshold value from the maximum tile average vegetation height ($\mathbf{h}$; m). The threshold logically determines which pairs of tiles are preemptively joined based on the absolute differences in their tile average vegetation heights ($\mathbf{\Delta H}$; m; Eqn 6; Figure S1).

$$\Delta\hbar_{n1,n2} < rht * \max(\mathbf{h}) \tag{6}$$

Second, the tile preservation parameter ($tpp$; number of tiles) prevents several tiles with the shortest average vegetation height from being merged. This means the tiling scheme will carry out preemptive joins based upon $rht$ or the default similarity criteria while preserving young recently disturbed tiles and explicitly representing early successional differences in fluxes (Bellassen et al., 2010; Zaehle et al., 2006; Nabel et al., 2020). When dynamic tiling is active, the time since disturbance is tracked in all tiles. Time since disturbance increases at the CTEM timestep (i.e. daily). Any disturbance events applied to a particular tile resets its time since disturbance to zero.





### 2.3.4 Externally forced fire

Externally forced fire builds upon the pre-existing fire module within CLASSIC (Melton and Arora, 2016; Arora and Melton, 2018). The annual fractional burned area in a grid cell is read from a file. The model assumes that fire impacts all non-crop PFTs.

If dynamic tiling is not active, biomass from the average individual and the litter pool burns proportional to the requested
fractional burned area. If dynamic tiling is active, a new tile with a fractional area equal to the fractional burned area is split from the active tiles within the grid cell and subsequently burned. Depending upon the requested fractional burned area and the conditions in the grid cell the model uses anywhere between one and the total number of tiles being actively simulated as candidate tiles for this splitting operation.

To determine the candidate tiles for this splitting operation, the model ranks the tiles based on their probability of fire ($\mathbf{p}$ of length $n$) conditional on the total aboveground biomasses available for burning ($\mathbf{b}$ of length $n$; kg C m$^{-2}$). $\mathbf{p}$ is a linear function of the lower biomass threshold (0.4 kg C m$^{-2}$) under which fire cannot sustain itself and the upper biomass threshold over which fire has a probability of one (1.2 kg C m$^{-2}$; Eqn. 7) (Moorcroft et al., 2001; Kucharik et al., 2000; Melton and Arora, 2016).

$$p_n = max\left[0, min\left(1, \frac{b_n - 0.4}{1.2 - 0.4}\right)\right] \tag{7}$$

The model initially selects tiles with a p of one as candidate tiles to be split to create the tile to be burned. However, if these selected tiles do not contain enough fractional area to simulate the fractional burned area requested, the model selects tiles with a p less than one from largest to smallest (Figure S1). Externally forced fire uses a single probability ($\mathbf{p}$) to rank tiles, whereas CLASSIC's standard fire module uses three probabilities to calculate the burned area: the probability of fire
conditional on total aboveground biomasses available for burning, the combustibility of the fuel based on its moisture content, and the presence of an ignition source (Arora and Boer, 2005; Arora and Melton, 2018). We make this simplification here because the fractional burned area comes from a file and all the tiles within a grid cell experience the same driving meteorology limiting differences in moisture content and ignition (Melton et al., 2017; Melton and Arora, 2014). With either dynamic tiling active or inactive, we calculate the C emissions to the atmosphere using pre-defined PFT-
specific fire emission fractions for each live vegetation component (i.e. both structural and non-structural leaves, stems, and roots) as well as the litter pool (υ; Table 1). We calculate the quantity of live vegetation C transferred to the litter pool as a result of fire-related mortality using pre-defined PFT-specific mortality fractions (Θ; Table 1). Externally forced fire does not impact crop PFTs and thus their biomass never combusts nor experiences fire-related mortality.



### 2.3.5 Externally forced harvest


Harvest simulates the removal of biomass from the landscape as a result of logging activities and builds upon the pre-existing land use change module within CLASSIC (Arora and Boer, 2010). The annual fractional harvested area on a per grid cell basis is read from a file. The model assumes that all harvest events are clear-cuts that impact some fraction of the simulated grid cell.


If dynamic tiling is not active, the average individual is harvested proportional to the requested fractional area. If dynamic tiling is active, a new tile with a fractional area equal to the requested fractional area is split from the oldest undisturbed active tile, and the entire newly created tile is harvested. If the harvested area requested exceeds the fractional area of the oldest undisturbed active tile the model selects additional active tiles as candidate tiles from oldest to youngest until there is

sufficient fractional area.

In either case, the harvested aboveground biomass (i.e. both non-structural and structural stem and leaf C) is split into three streams. These streams contribute C to the atmosphere, slash/pulp and paper products pool and durable wood products pool. The fractions of harvested aboveground biomass allocated to each stream ($\varepsilon$; Table 2) depend upon whether the PFT is woody or herbaceous and in the case of woody PFTs the aboveground biomass density (Arora and Boer, 2010). Unlike the

procedure described by Arora and Boer (2010) where root biomass is transferred to the slash/pulp and paper products pool, we transfer harvested root biomass to the applicable PFT and soil depth-specific litter pools.

### 2.4 Model forcing

CLASSIC requires seven meteorological forcing variables: incoming shortwave radiation, incoming longwave radiation, air temperature, precipitation rate, air pressure, specific humidity, and wind speed. We use the interpolated and disaggregated meteorological forcing described in detail by Meyer et al. (2021) and Curasi et al. (2022) (GSWP3–W5E5–ERA5) in our simulations. The 1901 – 1978 portion of the forcing comes from the Inter-Sectoral Impact Model Intercomparison Project GSWP3–W5E5 and the 1979–2018 portion comes from the ERA5 time series bias corrected to match the means of the

overlapping period in the GSWP3–W5E5 (Kim, 2017; Lange, 2019, 2020a, b; ECMWF, 2019). The atmospheric $CO_2$ concentrations (1700 - 2017) were obtained from the global carbon project (Trends in atmospheric carbon dioxide, National Oceanic & Atmospheric Administration, Earth System Research Laboratory (NOAA/ESRL), 2022; Friedlingstein et al., 2022). We prescribe the fractional coverage of PFTs using the remotely sensed 14 PFT-hybrid land cover product generated by Wang et al. (2022) and expanded upon and evaluated by Curasi et al. (2022). This land cover product combines

information from the North American Land Change Monitoring System land cover (Latifovic et al., 2017), the National Terrestrial Ecosystem Monitoring System (NTEMS)(Hermosilla et al., 2018, 2016), satellite-derived maps of the National



Forest Inventory attributes (Beaudoin et al., 2018), and British Columbia's biogeoclimatic ecosystem classification map (MacKenzie and Meidinger, 2018; Salkfield et al., 2016). Using a land cover that does not vary in time (i.e. prescribed land cover as opposed to dynamic land cover) allows us to focus on the influence of fire, harvest, and dynamic tiling on the model

outputs.

We develop fire and harvest drivers that detail the per-grid cell annual fractional area harvested or burned between 1700 and 2017 (Figure 2a, b). For the satellite era (1985 - 2017) we use remotely sensed 30-m spatial resolution records of harvest and fire events. These data were derived from Landsat by using breakpoint detection to identify changes and trends (Hermosilla

et al., 2016, 2015a), followed by a random forest classification of change types (Hermosilla et al., 2015b). We mask the remotely sensed harvest records to include only private, long-term tenure, and short-term tenure forests, as indicated by Stinson et al., (2019).

Before the availability of the remotely sensed records used herein (pre-1984), to our knowledge, there are no spatially

explicit pan-Canadian integrated harvest and fire data sets available. Therefore, to represent the impact of historical disturbance on the model state we employ established methods for inferring disturbance events from stand age for the period before reliable spatially explicit observations are available (Nabel et al., 2020; Kurz et al., 2009; Chen et al., 2000, 2003). We also focus our analysis of the CLASSIC simulations on the satellite era (1985 - 2017) due to uncertainties in inferred historical disturbance and the model state before the satellite era.


Maltman et al., (2023) derived a 30-m resolution stand age map for 2019 from Landsat and MODIS data utilizing three methods. The methods included disturbance detection for stands between 0 and 34 years of age, detection of spectral signals indicative of recovery for stands between 34 and 54 years of age, and inverting allometric equations for stands between 54 and 150 years of age. We infer the year in which the last disturbance occurred from the stand age. For example, a 40-year-

old forest in 2019 is assumed to have been last disturbed in 1979. We use regional averages of the per-pixel ratio of burned to total disturbed area from the first decades of the satellite era (1985 - 1995) to fraction total inferred disturbance into fire and harvest.

We utilized aspatial records of the total harvested and burned area within Canada to bias-correct inferred disturbance from

1920 - 1984 (Skakun et al., 2021; World Resources Institute, 2000). Before 1920, we utilized aspatial records of total disturbed area derived from 1920 stand age, with harvest held constant (0.3 Mha yr$^{-1}$) (Chen et al., 2000; Kurz et al., 1995). First, for years in which inferred burned or harvested area ($\mathbf{D}_{inferred}$ for $i$ years, $l$ grid cells; m$^2$) exceeded the aspatial records ($\mathbf{d}_{aspatial}$ for $i$ years; m$^2$) we correct the positive biases. We calculate an aspatial bias-correction factor ($f$ for i years; unitless; eqn. 8).



$$f_i = \frac{\left(\left(\sum_{l=1}^{n\,grid\,cells} d_{inferred,il}\right) - d_{aspatial,i}\right)}{\left(\sum_{l=1}^{n\,grid\,cells} d_{inferred,il}\right)}$$ (8)

Because the pre-1985 records are aspatial, the bias correction factor is temporally explicit but uniformly applied across space. When we apply the bias-correction factor to the inferred disturbance time series the result is a new time series with all the positive biases corrected ($\mathbf{D}_{downsc}$ for $i$ years, $l$ grid cells; $m^2$; eqn. 9).

$$d_{downsc,il} = d_{inferred,il} - d_{inferred,il}\, f_i$$ (9)

When we apply the bias-correction factor we retain the spatially and temporally explicit residuals ($\mathbf{D}_{residual}$ for $i$ years, $l$ grid cells; $m^2$; eqn. 10).

$$d_{residual,il} = d_{inferred,il} - d_{downsc,il}$$ (10)

Second, for years in which inferred burned or harvested area falls below the aspatial records, we correct the negative biases by adding in the residuals from nearby years (Figure 3). We loop backward in time from 1984 to 1740 to ensure that the

spatial and temporal patterns inferred from stand age are preserved during the negative bias correction. We accumulate residuals ($\mathbf{r}_{moving}$ for $l$ grid cells; $m^2$) extending as far back in time as needed to exceed the aspatial record for the year under consideration ($d_{aspatial,i}$). We calculate an aspatial bias-correction factor ($f$; unitless) and use it to apply a fraction of $\mathbf{r}_{moving}$, to the inferred disturbance time series and subtract the residuals used from $\mathbf{r}_{moving}$. When the spatially explicit residuals are exhausted (~1920 for fire only) they are replenished using the entire gridded remotely sensed and stand age inferred

disturbance record. This procedure continues until all the negative biases have been corrected between 1984 and 1740 yielding the final spatially explicit historical disturbance time series ($\mathbf{D}_{final}$ for $i$ years, $l$ grid cells; $m^2$).

## 2.5 Simulation protocol

We carry out a total of fourteen simulations using a common simulation protocol to investigate the impact of different

maximum numbers of available tiles, *rht*, and *tpp* (Table 3). *rht* (0.04 - 0.16, unitless) can be conceptually thought of as breaking the range of tile average vegetation heights into between 24 and 6 equally spaced bins depending upon its value (e.g. (1 bin m m$^{-1}$/ 6 bins) = 0.16 m m$^{-1}$). When *rht* is set these bins are used to group tiles to be preemptively joined. *tpp* (4 - 6 tiles) can be conceptually thought of as the maximum number of discrete disturbance events that the model can simulate in a grid cell over 2 - 3 years (e.g. (1 harvest tile yr$^{-1}$ + 1 fire tile yr$^{-1}$) * 3 yrs = 6 tiles). When *tpp* is set the model preserves

that number of recently disturbed tiles.

We spin up the model to equilibrium conditions corresponding to the year 1700 and then do a transient run over the period 1700 to 2017. For the spin up we loop the earliest 25 years of climate data available (1901 - 1925) and hold atmospheric $CO_2$ concentrations constant at the pre-industrial (1700) level. The 1700 - 1900 portion of the transient run uses the same loop of

1901 - 1925 climate, but transient atmospheric $CO_2$ concentrations. The 1900 - 2017 portion of the transient run uses



transient atmospheric CO₂ concentrations and evolving GSWP3–W5E5–ERA5 climate. During the full transient simulation from 1700 - 2017, the fire and harvest are applied (see Section 2.4). The CLASSIC nitrogen cycling module is not active in these simulations (Asaadi and Arora, 2021; Arora and Boer, 2010).

**2.6 Model evaluation**

We carry out model-on-model comparisons for a selection of variables and model configurations for the satellite era portions of our simulations (1985 - 2017) to select the optimal model setup that optimally balances detailed process representation and run time (Table 3). We also use these evaluations to demonstrate the relative impact of representing subgrid-scale heterogeneity within our modeling framework. We evaluate a suite of C cycling and surface energy balance-related variables including the land carbon pool (cLand), vegetation C (cVeg), soil C (cSoil), gross primary productivity (GPP), autotrophic

respiration (Ra), heterotrophic respiration (Rh), ecosystem respiration (ER), leaf area index (LAI), sensible heat flux (HFSS), latent heat flux (HFLS), albedo (ALBS), fire emissions (fFire), total deforested C (fDeforestTotal), and cumulative deforested C (the running sum of fDeforestTotal starting in 1985; fDeforestCumulative).

To select the optimal maximum number of tiles available for the simulation as well as the *rht* and *tpp* we calculate the mean

squared error (**mse** for *j* model runs over the 1985 - 2017 period) between the target model run and the reference 32-tile run. The 32-tile run is the reference point as it is the simulation with the least compromise between runtime and simulation detail and is assumed to best represent the impacts of disturbance in CLASSIC. mse is the mean of the squared differences between the annual summary (i.e. means for fluxes and sums for pools) of each variable for the 32-tile run ($\hat{x}_{32tile}$ containing *i* years) and that for the simulation under evaluation ($\hat{X}_{target}$ containing *i* years for *j* model runs; Eqn 11).

$$mse_{target,j} = \frac{1}{n\,years}\sum_{i=1985}^{2017}\left(\hat{x}_{target,ij} - \hat{x}_{32tile,i}\right)^2 \qquad (11)$$

We also use a normalized response metric ($\Delta\bar{X}_{norm}$ for, *j* model runs, *k* variables; Eqn 12) to evaluate the relative impacts of disturbance and subgrid-scale heterogeneity on the simulations and present a unitless summary of multiple model variables on the same axis. For a given variable, the metric normalizes each variable's output ($X_{target}$ for *i* years, *j* model runs, *k* variables, *l* grid cells) using the minimum and maximum across all the outputs ($X_{norm}$ for *i* years, *j* model runs, *k* variables, *l*

grid cells; Eqn 12). Each normalized variable is averaged across the model domain and run years ($\bar{X}_{norm}$ for, *j* model runs, *k* variables; 1985 - 2017) taking into account each model grid cell's area ($a_{grid\,cell}$; m²). Finally, the absolute value of the difference between $\bar{X}_{norm}$ for the target runs (1-tile/not-disturbed, and 32-tile run) and the 1-tile/disturbed run is calculated (Eqn 12).

$$x_{norm,i,j,k,l,m} = \frac{\left(x_{target,i,j,k,l,m} - \min\left(\boldsymbol{X_{target,k}}\right)\right)}{\left(\max\left(\boldsymbol{X_{target,k}}\right) - \min\left(\boldsymbol{X_{target,k}}\right)\right)}$$

$$\bar{x}_{norm,j,k} = \frac{\sum_{i=1985}^{2017}\left(\sum_{l=1}^{n\,grid\,cells} x_{norm,i,j,k,l,m}\, a_{grid\,cell,l}\right)}{33\left(\sum_{l=1}^{n\,grid\,cells} a_{grid\,cell,l}\right)}$$



$$\Delta \bar{x}_{norm,j,k} = \left| \bar{x}_{norm,j,k} - \bar{x}_{norm,1-tile/disturbed,k} \right| \tag{12}$$

All plots are created using R or the External Dynamic and Interactive Framework Integrating CLASSIC Experiments (EDIFICE) Python suite (Hijmans et al., 2015; R core team, 2013).

## 3. Results and discussion

### 3.1 Disturbance events within Canada

For the period represented by satellite data in this study, the highest total disturbed areas are in central boreal regions of the country and are attributable to wildfire events (Figure 2c). Alternately, harvest concentrates on the west coast and eastern boreal and maritime regions of the country. The annual total disturbed area differs widely between years during the satellite era (Figure 2a,b). In aggregate, harvest occurs in ~2% of the land area modeled, and fire occurs in ~6% from 1985 to 2017. The total number of simulated disturbance events is moderate with 89% of the grid cells incorporating 32 or fewer simulated disturbance events during the satellite era and 61% incorporating 11 or fewer (Figure 2d,e). The maximum number of possible events in the 1985 - 2017 drivers is 66 (i.e. a harvest and fire each year for 33 years; Figure 2e). Generally, beyond 11 simulated disturbance events, there is a limited correlation between the number of simulated disturbance events and the total area disturbed (Figure 2e).

### 3.2 Model parameterization

The change in the mse (Figures 4a-h) as the maximum number of available tiles for the run increases from 1 to 32 exhibits a roughly exponential decline for surface energy balance (HFSS, HFLS) and C cycle-related variables (cLand, GPP, ER, LAI). This reflects the model's ability to resolve the impact of subgrid-scale heterogeneity on these processes through increasingly complex simulations with fewer joined tiles. However, the impact of explicitly representing subgrid-scale heterogeneity saturates in increasingly complex simulations. This is likely because, as the number of tiles approaches the number of discrete disturbance events, the differences between the tiles being simulated become more nuanced and the statistical distributions of aboveground tree biomass in forested grid cells become increasingly similar (Figure 5a). The 32-tile simulation captures all the discrete disturbance events during the historical period across the majority of the model domain. We infer from the non-linear decreasing rate of change in Figures 4a-f that our reference 32-tile simulation approaches that of a computationally intractable simulation that resolves all disturbance events as tiles. The difference in mse between the 32-tile tile simulation and that computational intractable simulation would likely be vanishingly small similar to the difference between the 25-tile and 32-tile simulations (Nabel et al., 2020; Fisher et al., 2018; Ellner and Guckenheimer, 2011; Gelman and Hill, 2006).





Disturbance-related variables including fFire, as well as fDeforestedTotal exhibit a less sharp decline when going from one
to seven tiles and then approach an asymptote (Figures 4g-h). This reflects the role of selecting and splitting tiles in different
phases of recovery on these processes and the extent to which recovering tiles with lower aboveground tree biomass are
represented in the simulation (Figure 5a). As a result, there is a discontinuity between representing these processes using
average individuals, a small number of highly heterogeneous tiles, and a large number of tiles (Figures 4g-h, 5a). This
pattern is also likely influenced by the relatively low magnitude of the differences between the simulations when, compared
to the fluxes themselves (Figure 6a,h,i). The run time of the simulations increases linearly between one and twelve tiles
(Figure 4i) but increases more rapidly from eighteen to thirty-two tiles due to the increasingly large multidimensional per-tile
structures in memory. The intercept (i.e. the overhead for pre-processing meteorological files and initializing MPI sessions to
run the model with a single tile) is around three times the slope (i.e. the time required to run each additional tile) for
simulations with less than 12 tiles. This suggests that the splitting operations involved in simulating additional tiles are
computationally efficient and do not dramatically increase the run time (Nabel et al., 2020).

As with all modeling exercises, we must balance model accuracy, complexity, and computational efficiency. We, therefore,
use simulations with 12 tiles to set the *rht* (i.e. the relative height threshold for preemptively joining tiles) and *tpp* (i.e. the
number of preserved recently disturbed tiles) parameter values. Simulations using 12 tiles and different *rht* and *tpp*
parameters are very similar in terms of run time, and surface energy balance and C cycle-related variables exhibit no
consistent patterns (Figures S2a-f, i). There is a gradual increase in mse for fFire and fDeforestedTotal with higher *rht* and
*tpp* parameters (Figures S2g-h). However, these differences are again of relatively low magnitude, compared to the fluxes
themselves (Figure 6a,h,i). Therefore, we chose an *rht* of 0.16 and a *tpp* of 4 to maximize computation efficiency. The
optimal parameterization has mse values similar to those of a run with twelve tiles, but at a lower run time (~23% less;
Figure 4i). The optimal model nearly approximates the heterogeneous tile structure of the more complex 32-tile simulation
and represents forested areas with low aboveground tree biomass similar to the 32-tile simulation, in line with observations
of aboveground tree biomass from the NFI (Figure 5b). However, it may over smooth the transition between low and high
biomass areas (i.e. the ~2 - 3 gC m$^{-2}$ range in Figure 5b) thereby impacting the size classes of the tiles selected for splitting
during the disturbance simulation (Figure 4g-h). Nonetheless, the optimal simulation effectively balances computational
efficiency, and discretization error (Nabel et al., 2020; Fisher et al., 2018).

### 3.3 Impacts on simulated variables

The response of the modeled variables to dynamic tiling (i.e. 1-tile/disturbed v.s. 32-tile, as illustrated in Figure 1, Table 3)
often meets or exceeds their response to disturbance alone (i.e. 1-tile/not-disturbed vs. 1-tile/disturbed; Figure 6a). The
impact of the optimal tiling scheme is minimal by comparison and, therefore, we focus on comparisons between the 1-
tile/not-disturbed, 1-tile/disturbed, and 32-tile outputs (Figures 6b-i, S3). C cycle-related variables including cVeg, and LAI
show the strongest response to disturbance, whereas energy balance-related variables including HFLS, HFSS, and ALBS





show the weaker responses. Many variables also respond strongly to dynamic tiling including LAI, Ra, and cVeg (Figure 6a,b,e). Select surface energy balance-related variables including HFLS, HFSS, and ALBS respond more strongly to dynamic tiling than disturbance alone (Figure 6a). These strong responses further reinforce the impact of disturbance-induced subgrid-scale heterogeneity on ecosystem processes and the value of representing this heterogeneity within models (Bellassen et al., 2010; Zaehle et al., 2006; Nabel et al., 2020; Körner, 2006; Dore et al., 2010; Luyssaert et al., 2014; Erb et al., 2017).

Disturbance-related variables such as fFire exhibit little difference with subgrid-scale heterogeneity (Figure 6h), whereas fDeforestCumulative increases slightly (Figure 6i). These patterns occur as fire can potentially impact all subgrid stands above a certain biomass threshold (Eqn. 7) while wood harvest preferentially impacts the tiles with the largest aboveground biomass (i.e. approximating the highest quality tiles being harvested). Biomass removal by disturbance leads to a ~1.6 Pg decrease in cVeg across Canada (an 8% decrease) while the subgrid level (tiled) representation of these processes leads to another ~1 Pg decrease (Figure 6b). LAI mirrors these same patterns with a 4% decrease due to disturbance and another 4% decrease with the subgrid level (tiled) representation (Figure 6e). As a result of disturbance, cSoil decreases at a higher rate from 1985 to 2017, whereas cVeg increases at very similar rates ~0.035 Pg $y^{r-1}$ (Figure 6b,c). For GPP and Ra, the impact of dynamic tiling is ~1.5 - 2.5 times the impact of disturbance alone (Figure 6a,d). This offset in GPP and Ra is likely in part a product of dynamic tiling simulating the naturally slower process of recovery from bare ground versus recovery of an average individual with substantial pre-existing biomass and leaf area (Zaehle et al., 2006; Körner, 2006; Dore et al., 2010; Luyssaert et al., 2014). This slower recovery likely also contributes to the losses in cVeg between the 1-tile/disturbed and 32-tile simulation.

Dynamic tiling has impacts on HFLS, HFSS, and ALBS ~1.5 - 4 times the impact of disturbance alone. The relative impact on HFLS is the more muted of the three possibly because while removing vegetation causes a decrease in transpiration, evaporation from the ground surface increases (Figure 6g). Dynamic tiling does appear to have larger impacts on ALBS where the surface becomes brighter and by extension, HFSS decreases (Figure 6a,f). This occurs as the model with dynamic tiling is capable of representing sparsely vegetated and often snow-covered fractions of the land surface as a result of recent disturbance (Bright et al., 2013; Nabel et al., 2020). These impacts are muted or absent when disturbance is simulated by an average individual model because only a proportion of the average individual's biomass is removed with enough remaining to support a tall dense canopy that obscures the ground surface and recovers quickly.

### 3.4 Implications for representing disturbance and subgrid-scale heterogeneity in LSMs

The dynamic tiling scheme presented in this study could form the basis for a more detailed representation of land use change and resultant subgrid-scale heterogeneity in CLASSIC, the land surface component of the LSM CanESM (Melton et al., 2020; Seiler et al., 2021; Swart et al., 2019). Our tiling scheme has several advantages over other methods. It uses a



relatively large number of dynamic tiles as opposed to a small fixed number of tiles, which allows for a more granular representation of vegetation recovery following disturbance (Shevliakova et al., 2009; Yue et al., 2018a; Naudts et al., 2015; Yang et al., 2010; Stocker et al., 2014). It also explicitly simulates C and energy exchanges using tile average rather than grid cell average properties thus fully simulating the impacts from the removal of vegetation by harvest or fire (Bellassen et al., 2010; Haverd et al., 2014; Melton and Arora, 2014). Most importantly, the scheme is dynamic and has no designated size or age class bins; the number of simulated tiles increases as disturbances occur and then are managed by on-demand or pre-emptive joins (Nabel et al., 2020; Shevliakova et al., 2009; Naudts et al., 2015; Bellassen et al., 2010). The tiling routine adapts its size distribution in response to lower disturbance frequencies, more extreme individual disturbance events, and potentially the addition of new PFTs while remaining computationally efficient. Finally, the tiling scheme can preserve young, recently disturbed tiles which may improve its representation of early successional differences in GPP, LAI, and cVeg (Bellassen et al., 2010; Zaehle et al., 2006; Nabel et al., 2020). In the context of pan-Canadian or global offline simulations within CLASSIC, this dynamic tiling scheme presents the opportunity for more detailed and efficient representations of LULCC than is achievable by simply increasing the spatial resolution of the model, which is limited by model inputs such as meteorological forcing or coupling considerations within CanESM. Future LULCC representations could implement more complex tile harvesting schemes to represent forest management (i.e. thinning, re-planting, clearcut avoidance, or low-intensity harvest) (Puettmann et al., 2015; Pan et al., 2010) or introduce tiles to account for new LULCC processes and states such as rangelands, pasture, fertilizer use, and irrigation (Shevliakova et al., 2009).

Finally, our model-on-model evaluation provides insights into the biases induced in specific variables by the absence of dynamic tiling or particular dynamic tiling setups, which may apply to other similar tile-based LSMs/discretized schemes (Nabel et al., 2020; Fisher et al., 2018). Representing subgrid-scale stand structure leads to differences in land-use emissions if particular size or age classes within the grid cell are preferentially impacted by fire or harvest (Nabel et al., 2020; Shevliakova et al., 2009; Yue et al., 2018b). Our results suggest that representing a relatively small number of heterogeneous tiles (e.g. < 12) may yield undesirable biases when compared to simulations using a larger number of tiles (32-tile; Figures 4a-h, 6a) (Yue et al., 2018a; Shevliakova et al., 2009; Stocker et al., 2014; Yang et al., 2010). For a tile-based LSM to represent these subgrid impacts the simulation needs to be sufficiently complex (12+ tiles in the case of CLASSIC) and judiciously implemented and tested. Likewise, subgrid-scale stand structure impacts C fluxes, vegetation C stocks, and energy fluxes (Erb et al., 2017; Luyssaert et al., 2014; Körner, 2006; Dore et al., 2010). These subgrid-scale impacts can be of similar magnitude to the impacts of disturbance alone, further reinforcing their significance (Figure 6).

## 4. Conclusions

Canadian forest ecosystems are critical components of the global C cycle which are responding to unprecedented climate change. We develop an optimal parameterization for fire and harvest tailored to Canada which also represents the subgrid-

scale heterogeneity resulting from disturbance. We demonstrate that representing this subgrid-scale heterogeneity has impacts on grid-scale vegetation, C stocks, and C fluxes and to a lesser extent the surface energy balance-related variables above and beyond the disturbance impacts themselves. Our approach to dynamically represent subgrid-scale heterogeneity using tiles may apply to other tile-based LSMs. Ultimately, quantifying historical disturbances and evaluating the impacts of different methods of representing disturbance will improve the representation of the terrestrial C cycle in LSMs. This understanding will also facilitate a comprehensive, process-based assessment of Canada's future terrestrial C cycle and its response to both disturbance events and climate change.

**Code and data availability**

The Canadian Forest Service land cover and maps of forest disturbance described herein for Canada's forested ecosystems are open access and are freely available at https://opendata.nfis.org/mapserver/nfis-change_eng.html. The Stinson et al., (2019) forest management product is available through the Government of Canada's Open Data Portal (https://open.canada.ca/data/en/dataset/d8fa9a38-c4df-442a-8319-9bbcbdc29060). The current version of CLASSIC is available via the project website: https://gitlab.com/cccma/classic. The model version, additional software, setup files, and outputs used herein are archived on Zenodo (https://doi.org/10.5281/zenodo.8302974).

**Author contributions**

S.R.C., J.R.M., and E.R.H. conceived of the analysis and methodology. S.R.C. conducted the formal analysis, visualization, and software development, and wrote the original draft. J.R.M. and E.R.H. obtained funding for and oversaw the work. T.H. and M.A.W. contributed the National Terrestrial Ecosystem Monitoring System data and to the disturbance-forcing development. All authors contributed to writing and editing the manuscript.

**Competing interests**

The authors declare that they have no conflict of interest.

**Acknowledgments**

We would like to thank Mike Brady (ECCC) for his assistance in processing the disturbance data, Ed Chan for his assistance with the model software and setting up the original model initialization files and meteorological drivers, and Libo Wang for creating the original land cover products and cross-walking tables. We acknowledge the support of the Natural Sciences and Engineering Research Council of Canada (NSERC), ALLRP 556430-2020. We also thank the anonymous reviewers for their constructive comments.



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



## Figures

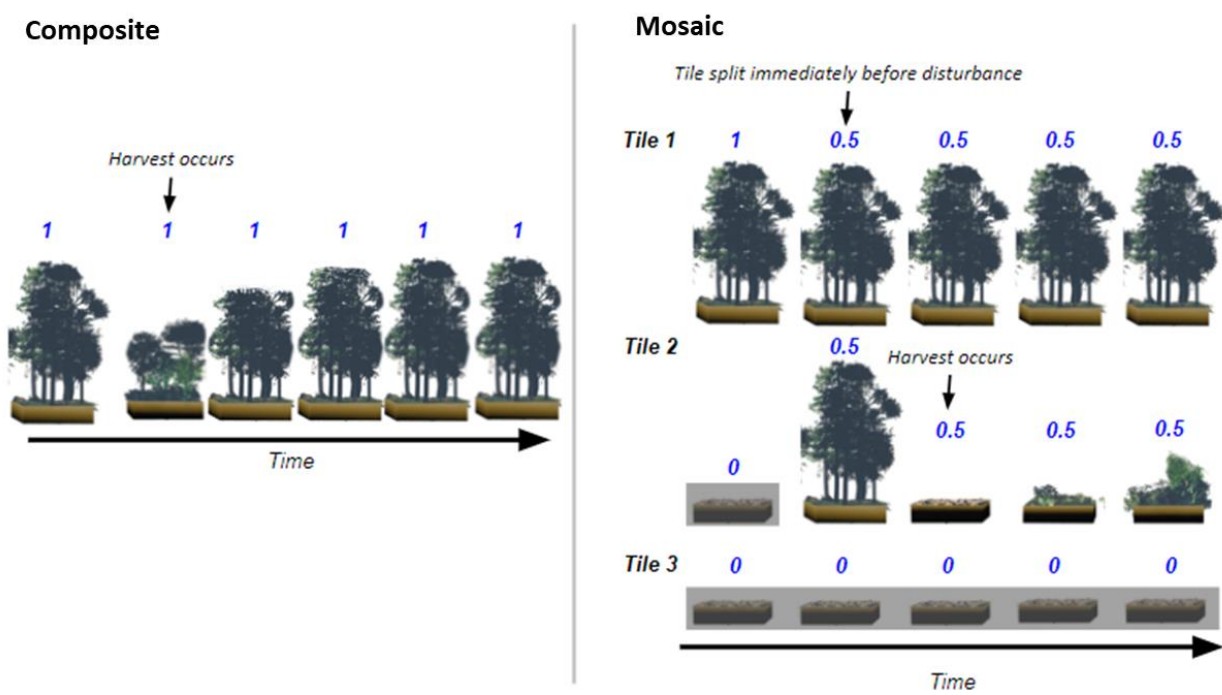


**Figure 1: Illustrative diagram contrasting the composite (1-tile) and mosaic (>1 tile) representations of disturbance implemented herein assuming a hypothetical scenario where 50% of a grid cell undergoes timber harvest. The fraction of a grid cell that is occupied by a tile is denoted above each tile. Tiles that are not active have a gray background (e.g. Tile 3 across all time steps).**




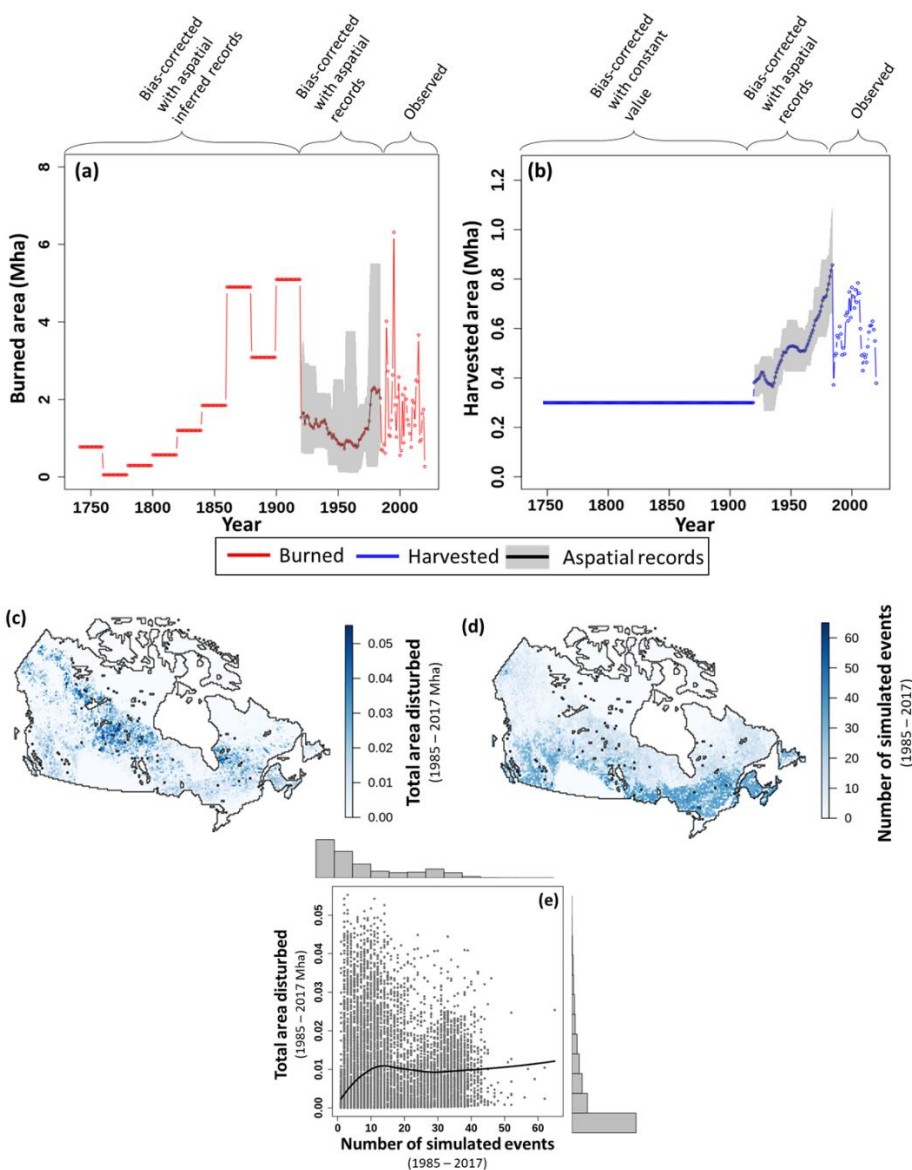

**Figure 2: Plots of the disturbance drivers over time. a) annual total burned and b) harvested area from 1740 - 2020. Observed indicates the period that uses the Landsat fire and harvest observations (Hermosilla et al., 2016, 2015a, b). Bias-corrected with aspatial records indicates the period where the disturbance was inferred from 2019 stand age (Maltman et al., 2023) and bias-corrected using aspatial harvested and burned area (Skakun et al., 2021; World Resources Institute, 2000; Van Wagner, 1988). Bias-corrected with aspatial inferred records and bias-corrected with constant value indicate the period where the inferred disturbance was bias-corrected based on Kurz et al., (1995) and Chen et al., (2000) respectively. The aspatial records line is the nine-year running mean, min., and max. of the aspatial total harvested and burned area data sets. c) Per-grid cell total area disturbed (1985 – 2017) and d) the total number of simulated events (1985 - 2017). e) Per-grid cell total area disturbed, excluding un-disturbed cells, plotted against the total number of simulated events. The black line is a LOESS curve. Note that a simulated event combines all the individual fire or harvest events that occur in a grid cell in a single year, with a maximum of two simulated events per year (one fire and one harvest) occurring in each grid cell.**





Start ($i = 1985$; $j = 1984$; $r_{moving,l} = 0$)

Move to the preceding time step
$i = i - 1$

Is negative bias correction required?

$$\left(\sum_{l=1}^{n\ grid\ cells} d_{downsc,il}\right) < d_{aspatial,i}$$

**No**

**Yes**

Are there sufficient residuals?

$$\left(\sum_{l=1}^{n\ grid\ cells} r_{moving,l}\right) \geq d_{aspatial,\ i}$$

**No**

Add in a preceding year's of residuals[1]

$$r_{moving,l} = r_{moving,l} + d_{residual,jl}$$

$$j = j - 1$$

**Yes**

Calculate and apply the bias correction

$$f = \frac{\left(d_{aspatial,i} - \left(\sum_{l=1}^{n\ grid\ cells} d_{downsc,il}\right)\right)}{\left(\sum_{l=1}^{n\ grid\ cells} r_{moving,l}\right)}$$

$$d_{final,il} = d_{downsc,il} + r_{moving,l}\ f$$

Subtract the residuals used

$$r_{moving,l} = r_{moving,l} - r_{moving,l}\ f$$

Have all the years been corrected?

$$i == 1740$$

**No**

**Yes**

End

Key:

Logical decisions

Calculations/operations

**Figure 3: A schematic diagram of the negative bias correction algorithm with applicable equations and logical tests. [1]Not shown: When the spatially explicit residuals are exhausted they are replenished using the entire remotely sensed and stand age inferred disturbance record.**






**Figure 4: Plots of the mean squared error (mse; 1985 – 2017; Eqn 11) for a) the land carbon pool (cLand), b) gross primary productivity (GPP), c) ecosystem respiration (ER), d) leaf area index (LAI), e) sensible heat flux (HFSS), f) latent heat flux (HFSS), g) fire emissions (fFire), and h) total deforested carbon (fDeforestTotal) for model runs including disturbance with varying numbers of tiles (1 – 32) compared against the run including disturbance with the largest number of tiles (32). i) The run time for each configuration. Values are also shown for the optimal model run including disturbance with 12 tiles (tile preservation parameter [*tpp*] = 4; and relative height threshold [*rht*] = 0.16).**





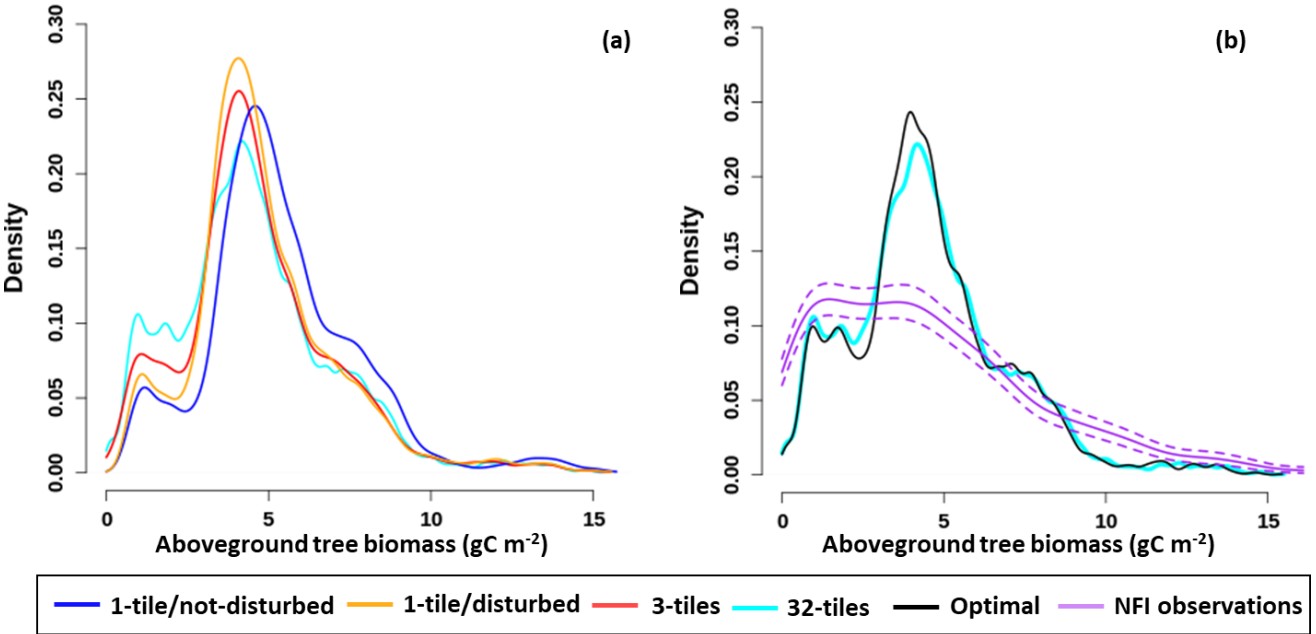

**Figure 5: Weighted histogram of aboveground tree biomass for forested areas of Canada at the end of a) a selection of model runs including the 1-tile/not-disturbed run, 1-tile/disturbed, 3-tiles, and 32-tiles. As well as for b) 32-tile, optimal (12 tiles, 4 preserved tiles, and a threshold of 0.16), and observations from the National Forest Inventory (NFI) (Gillis et al., 2005). All runs using >1 tile include disturbance. The bootstrapped 95% CI for the NFI observations is also shown. The contributions of all forested subgrid areas weighted by their fractional area within the modeled region are considered. An area is classified as forested if it contains at least 50% tree cover.**







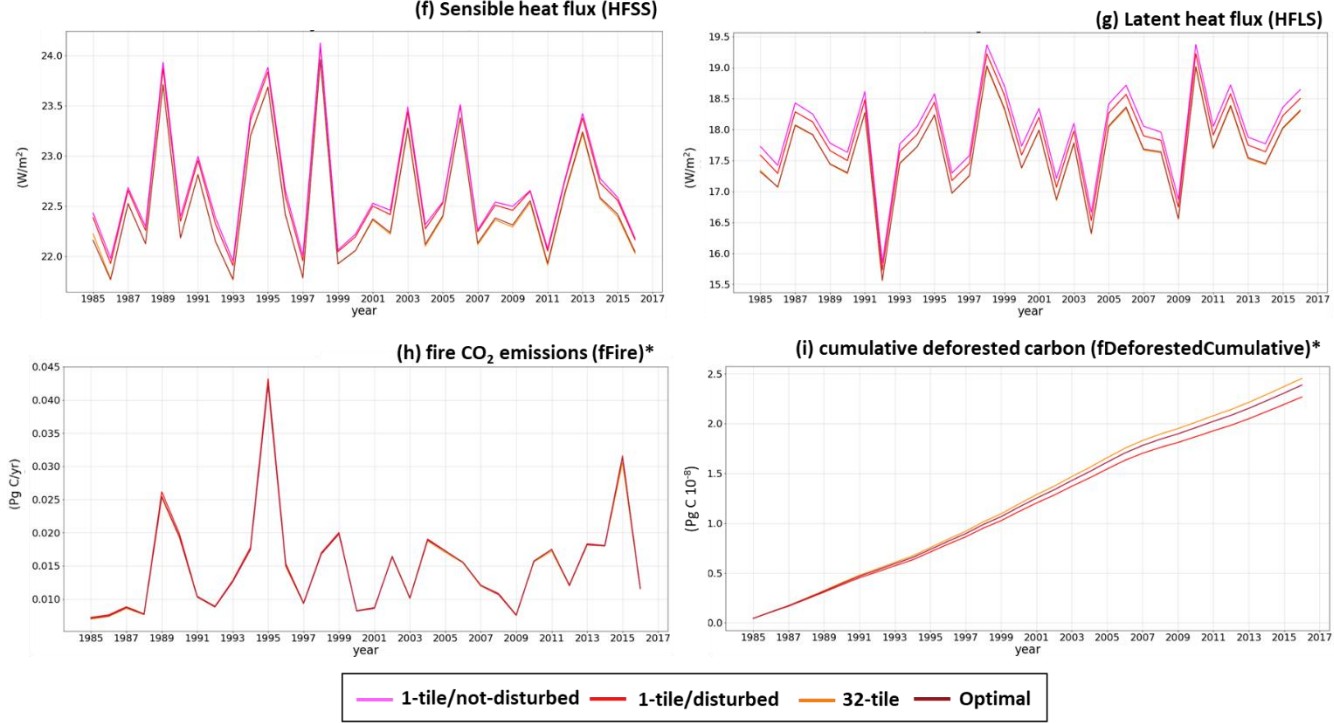


**Figure 6: a) Plot of the normalized response metric (ΔX̄norm; Eqn 12) for the 1-tile/not-disturbed versus 1-tile/disturbed and 1-tile/disturbed versus 32-tile for vegetation carbon (cVeg), soil carbon (cSoil), gross primary productivity (GPP), autotrophic respiration (Ra), heterotrophic respiration (Rh), leaf area index (LAI), sensible heat flux (HFSS), latent heat flux (HFLS), albedo (ALBS), fire emissions (fFire), and total deforested carbon (fDeforestTotal). Time series plots of b) cVeg, c) cSoil, d) GPP, e) LAI f)**
**HFSS, g) HFLS, h) fFire, and i) cumulative deforested carbon (fDeforestCumulative; the running sum of fDeforestTotal starting in 1985) for 1-tile/not-disturbed, the 1-tile/disturbed, 32-tile and optimal (*tpp* = 4; *rht* = 0.16; purple). All runs using >1 tile include disturbance. In the "a" panel a normalized response of zero indicates that there are no differences between the runs. *denotes disturbance-related fluxes that are omitted in the 1-tile/not-disturbed model run.**

**Tables**

**Table 1: The PFT-specific fire emission fractions (υ) used to calculate C emissions to the atmosphere due to fire for each live vegetation component (i.e. both structural and non-structural leaves, stems, and roots) and the litter pool. As well as the PFT-specific mortality fractions (Θ) used to calculate the quantity of C from each live vegetation component transferred to the litter pool. Crop PFTs are not impacted by fire and therefore not assigned fractions (Melton and Arora, 2016).**

| PFT type | green leaves combusted ($υ_L$) | green leaves litter ($Θ_L$) | brown leaves combusted ($υ_B$) | brown leaves litter ($Θ_B$) | stems combusted ($υ_S$) | stems litter ($Θ_S$) | roots combusted ($υ_R$) | roots litter ($Θ_R$) | litter combusted ($υ_D$) |
|---|---|---|---|---|---|---|---|---|---|
| Tree | 0.42 | 0.20 | - | - | 0.12 | 0.60 | 0.00 | 0.10 | 0.30 |
| Herbaceous | 0.48 | 0.10 | 0.54 | 0.06 | 0.00 | 0.00 | 0.00 | 0.25 | 0.42 |
| Shrub | 0.42 | 0.20 | - | - | 0.12 | 0.60 | 0.00 | 0.10 | 0.36 |




**Table 2: The fraction of harvest-affected biomass transferred to different wood product pools for herbaceous PFTs and woody PFTs (ε). The fractions for woody PFTs differ depending on aboveground biomass density (Arora and Boer, 2010).**

|  | Aboveground biomass density (kgC m$^{-2}$) | Fraction of deforested biomass emitted to the atmosphere ($\varepsilon_A$) | Fraction of deforested biomass as slash/pulp and paper products ($\varepsilon_D$) | Fraction of deforested biomass as durable wood products ($\varepsilon_S$) |
|---|---|---|---|---|
| Woody PFTs | > 4.0 | 0.15 | 0.70 | 0.15 |
|  | 1.0 - 4.0 | 0.30 | 0.70 | 0.00 |
|  | < 1.0 | 0.45 | 0.55 | 0.00 |
| Herbaceous PFTs | NA | 0.45 | 0.55 | 0.00 |

**Table 3: An overview of the simulations conducted in this study.**

| Abbreviation | Land surface representation | Includes disturbance | Max available tiles | Relative height threshold | Tile preservation parameter |
|---|---|---|---|---|---|
| 1-tile/not-distubed | composite | No | 1 | - | - |
| 1-tile/disturbed | composite | Yes | 1 | - | - |
| 3-tile | mosaic | Yes | 3 | - | - |
| 7-tiles | mosaic | Yes | 7 | - | - |
| 12-tile | mosaic | Yes | 12 | - | - |
| 18-tile | mosaic | Yes | 18 | - | - |
| 25-tile | mosaic | Yes | 25 | - | - |
| 33-tile | mosaic | Yes | 33 | - | - |
| Optimized | mosaic | Yes | 12 | 0.04 | 4 |
| - | mosaic | Yes | 12 | 0.08 | 4 |
| - | mosaic | Yes | 12 | 0.16 | 4 |
| - | mosaic | Yes | 12 | 0.04 | 6 |
| - | mosaic | Yes | 12 | 0.08 | 6 |
| - | mosaic | Yes | 12 | 0.16 | 6 |
