# Peer review of "Implementing a dynamic representation of fire and harvest including"

_EGUsphere, 2023_

## Referee Comment (RC1)

**Review of "Implementing a dynamic representation of fire and harvest including subgrid-scale heterogeneity in the tile-based land surface model CLASSIC v1.45"**

by Curasi et al.
EGUsphere 2023-2003 for GMD

Curasi and colleagues present an implementation of sub-gridcell heterogeneity in the CLASSIC LSM, where the subgridcell tiles are dynamically created when the model simulates fire or harvest events. They test the model over Canada (offline with no coupling to an atmospheric model) with a custom set of PFTs, and find that the sub-gridcell heterogeneity makes as much, in fact often larger difference, to certain important C pool and energy flux variables than disturbance.

The presented advance is appropriate for publication in GMD. The topic is relevant for land surface modelling and therefore earth system modelling in general. The details of the implementation seem sensible, and the critical examination of the number of tiles with respect to balancing "accuracy" (the quotes because they don't compare to data, just to the "full" model) and run time is very welcome. The effect on the fluxes and pools of the disturbances and the subgridcell heterogeneity are not negligible (so probably worthwhile enabling in many simulation setups), but not huge either (it will be interesting to see what affect they have in global, coupled simulations). I do have a few points of criticism which I believe should be addressed. Some of there are rather important to my mind, so I am going to recommend "Major Revisions" – even though the criticism may be considered comparatively mild for such a recommendation.

Key issues (in rough order of importance):

1. The authors made a commendable effort in setting up pre-satellite era disturbance scenarios for their model simulations to ensure a reasonable initial state before the evaluation period. But the time series for harvested area (Fig 2b) shows a huge discontinuity at the transition from bias corrected to observed. This is very probably not correct unless by some massive coincidence there was a change in legislation around that time or a bunch of logging companies simultaneously went bust. More likely the bias-correction procedure has gone wrong. Looking as the plot, the underlying data appear fine as the rate of increase is broadly similar across both periods. So if the earlier period was simply shifted downwards, the trajectory would be eminently reasonable! This should be investigated and fixed. The good news is that it will likely might make the simulated effects stronger because less logging beforehand will leave larger biomass pools at the start of the evaluation period, so the effects of disturbance and sub-gridcell heterogeneity will likely be stronger.

2. It is not made explicitly clear that when the number of tiles is greater than the number of disturbance events in a gridcell, the results for the gridcell will not improve (or change at all) with allowing more tiles (because they won't be used). I assume the authors understand this, but the closest that the manuscript says to this is lines 426-433, although they don't say this clearly. Instead the text talks about a "roughly exponential decline" and mentions "saturation". This text is very wordy (it is hard to read) and misses the most important point: when you increase the number of dynamics tiles to 7, the disturbances in most gridcells are perfectly (or near perfectly) resolved and the results agree with the 32 tile run (for most variables). This should be clearly mentioned and its impact on the conclusions better discussed.

Relatedly, I need to flag the statement in the Discussion (line 528):

"Our results suggest that representing a relatively small number of heterogeneous tiles (e.g. < 12) may yield undesirable biases when compared to simulations using a larger number of tiles (32-tile; Figures 4a-h, 6a)".

Based on Fig 4, the 7-tile simulation pretty much nails the 32-patch run and so doesn't yield "undesirable biases". The exception is the fire $CO_2$ emissions but these emissions are very insensitive to the tiling (Fig 6h). Also, Fig 6a) only shows comparisons of 1 vs 32 tiles, values like 7 and 12 are not shown, it is cannot support the statement.

Based on the two points above, I think the discussion of optimal number of tiles needs to be reconsidered.

3. The manuscript is rather long, but despite all the words, the wording is not always clear and can be difficult to follow. Here is an example starting at line 344:

"We utilized aspatial records of the total harvested and burned area within Canada to bias-correct inferred disturbance from 1920 - 1984 (Skakun et al., 2021; World Resources Institute, 2000). Before 1920, we utilized aspatial records of total disturbed area derived from 1920 stand age, with harvest held constant (0.3 Mha yr -1 ) (Chen et al., 2000; Kurz et al., 1995). First, for years in which inferred burned or harvested area (D inferred for i years, l grid cells; m 2 ) exceeded the aspatial records (d aspatial for i years; m 2 ) we correct the positive biases."

Problem I encountered when reading this:

a. The order of the first two sentences should be swapped because they don't match the conceptual temporal ordering (i.e. should be pre-1920 and *then* 1920-1984). In fact, both sentences could be combined into something much more succinct since they repeat a lot of words and deal very much with the same topic.
b It is not immediately clear if the bias correction was also applied to the pre-1920 data or not despite the data getting its own sentence. Likely not, but the sentence describing this data comes in between the first mention of the bias-correction and the description of it, so the implication is that it was in fact applied to pre-1920 data…? I don't know what to make of it.
c. Getting into the description (third sentence), what is the "inferred" burned area or harvested area?
d. Why are positive and negative biases being corrected differently? No explanation is given.
e. In general, what follows this text is a (confusing) technical explanation of the bias correction procedure . At no point were the goals of the bias correction mentioned, potential pitfalls, the rationale for the choices made etc. And what is this "loop backward in time"? This is a completely new idea to me, why was this done?

As mentioned above, another example can be found in lines 426-432. Many words are used to basically say "When there are more tiles than disturbances, the results won't change if you add more tiles."

These are just two particular examples. The text is afflicted throughout with poor and non-logical flow, awkward wording and inappropriate levels of detail.

**Because of the above, large parts of the manuscript should be critically reviewed and re-written.**

Some general suggestions:

- Use more subheadings. This will force the writer to focus more precisely on what that text is supposed to say and will help the reader to understand exactly what they should be taking from the text.
- Describe the only methods briefly in the main text, but move the details to the appendix. This will improve the flow significantly.
- One specific point, please include more high-level details of the "normalized response metric". How should the values be interpreted its their scale? And maybe give one sentence summarising the construction of the variable before jumping in the technical details.

4. The overall quality of the language is not too bad in terms of grammar and style, but it needs tightening up. Examples (non-exhaustive):

- Line 125 - "We use a domain, which encompasses all of Canada south of 76°N as our study area for demonstration." Technically correct but very awkward .
- Line 191-195 – This single sentence sentence is huge (fully four lines with not a single comma) and completely cryptic. Reformulate.
- Line 236-238 – Not really a proper sentence.
- Line 417 – "Alternately" is not what the authors mean. I think "In contrast" or "Contrastingly".
- Line 466 – Why are there citations behind an assertion about the paper's own results?
- Line 524 – Use of the term "biases". This is a matter of taste, but I dislike using "bias" when referring to model-to-model comparisons and prefer to reserve it for model-to-data comparisons. The authors may want to consider a different term.

5. There is very little comparison of the model improvements to data (only Fig 5b), but a large amount of model-to-model comparisons. The stated logic is that the 32-tile simulation with disturbance enabled is the best possible simulation, and therefore the standard to which the other simulations should be compared. But, is this 32-tile, disturbance-enabled simulation actually any better when compared to data than a single-tiled, non-disturbed simulation? Logically it might be better, but highly generalised but complex process-based models (such as LSMs) might not respond the way one expects. Some further model-to-data comparisons should be considered to make sure that changes are not actually making the model skill worse due to pre-existing cancellation of errors or some bias.

Relatedly, the explantation of the pre-existing "mosiac" tiling feature is not really clear (line 171). It is not specified how the tiles differ from one another – one must assume it varies across the cited studies. Giving more details about these studies and briefly summarising how mosaic tiling improved the model compared to the default "composite" approach would help justify the near exclusive use of model-to-model comparisons in the present study.

6. The authors made a point of describing the max of 66 disturbance events (due to the 33 years of the evaluation period) and relating that to the number of tiles. Simultaneously they spend a lot of effect deriving historical disturbance scenarios and also clearly state that disturbance is applied throughout. But they don't make it clear how the disturbance is applied in the pre-evaluation period (since the can't be contributing to the maximum 66 events). After carefully re-reading, it seems likely that the *composite* representation was used in the earlier period but I don't think this is stated anywhere. If it was, will this transition from composite to mosaic change effect the results, particularly in the early phase of the evaluation period? Please make clear and discuss.

7.  There is no mention of other disturbance agents such as biotic agents (importantly bark beetle), drought, wind throw and land slips.  It is fine that they weren't included (not everything can be modelled at once) but some discussion is needed.  Why where certain disturbances chosen and not others? Could this approach apply to other disturbance agents? What do the authors believe the consequences of leaving these out could be?

---

## Author Comment (AC1)

Dear Editor,

Please find our reply to Reviewer #1 below. We copy the review in full and our replies are in **bold blue**.

*Reviewer #1*

Curasi and colleagues present an implementation of subgrid-cell heterogeneity in the CLASSIC LSM, where the subgrid-cell tiles are dynamically created when the model simulates fire or harvest events. They test the model over Canada (offline with no coupling to an atmospheric model) with a custom set of PFTs, and find that the sub-gridcell heterogeneity makes as much, in fact often larger difference, to certain important C pool and energy flux variables than disturbance. The presented advance is appropriate for publication in GMD. The topic is relevant for land surface modeling and therefore earth system modeling in general. The details of the implementation seem sensible, and the critical examination of the number of tiles with respect to balancing "accuracy" (the quotes because they don't compare to data, just to the "full" model) and run time is very welcome. The effect on the fluxes and pools of the disturbances and the subgrid cell heterogeneity are not negligible (so probably worthwhile enabling in many simulation setups), but not huge either (it will be interesting to see what effect they have in global, coupled simulations). I do have a few points of criticism which I believe should be addressed. Some of these are rather important to my mind, so I am going to recommend "Major Revisions" – even though the criticism may be considered comparatively mild for such a recommendation.

**Thank you for volunteering to review our article and provide your comments. We appreciate your recognition of the value of this work. We also appreciate your review of the manuscript and the opportunity to further improve the text. Below are our responses to your comments alongside descriptions of our likely edits to the text.**

Key issues (in rough order of importance):

1. The authors made a commendable effort in setting up pre-satellite era disturbance scenarios for their model simulations to ensure a reasonable initial state before the evaluation period. But the time series for harvested area (Fig 2b) shows a huge discontinuity at the transition from bias corrected to observed. This is very probably not correct unless, by some massive coincidence, there was a change in legislation around that time or a bunch of logging companies simultaneously went bust. More likely the bias-correction procedure has gone wrong. Looking at the plot, the underlying data appear fine as the rate of increase is broadly similar across both periods. So if the earlier period was simply shifted downwards, the trajectory would be eminently reasonable! This should be investigated and fixed. The good news is that it will likely might make the simulated effects stronger because less logging beforehand will leave larger biomass pools at the start of the evaluation period, so the effects of disturbance and subgrid-cell heterogeneity will likely be stronger.

**Reconstructing the trajectory of historical fire and harvest within Canada is a difficult problem, due to the limited spatially explicit historical records. We chose to reconstruct the historical trajectory of disturbance using the best data sources available and explicitly highlight the methodology and results therein rather than conducting our model-on-model experiments using idealized scenarios (i.e. the average of the observed applied uniformly**

**across space). We rely on a mix of stand age, and aspatial historical records to carry out our reconstruction. We treat the historical records available to us for given periods as authoritative and accurate. The harvest information available for Canada (Figure 6 in World Resources Institute 2000) ranges from ~0.4 Mha in 1920 to ~0.9 Mha in 1985 matching what is shown in figure 2b. Therefore, this discontinuity is due to the transition from aspatial records to satellite-based observations, rather than a technical issue in the bias correction. There is work ongoing to unify a wider range of disparate spatial and aspatial historical records and reconstruct the trajectory of fire and harvest over time. Such further developments will be key to deploying this model to simulate the historical trajectory of the Canadian carbon cycle.**

**These challenges are further mitigated by the questions addressed in the manuscript and the types of experiments carried out. Our goals herein are to demonstrate the impacts of representing disturbance and sub-grid scale heterogeneity on carbon and energy fluxes and the trade-offs involved in incorporating more detailed process representation (i.e. more tiles). Therefore, we conducted model-on-model evaluations. All our simulations (for example shown in figure 4) utilize the same fire and harvest trajectory and we focus on the period from 1985 to 2017 when the model is driven by observations. Moreover, in figure 6 rather than evaluate the impact of tiling in absolute terms we evaluate it relative to the impact of disturbance. Finally, harvest represents a relatively small fraction of the pre-1985 disturbed area when compared to fire. Overall, our goal in constructing this analysis was that any modifications to the pre-1985 fire and harvest would impact all the simulations therein. This leads to conclusions that are robust to the drivers and address broader questions related to the representation of sub-grid scale heterogeneity in models, rather than the specific case and trajectory of Canada. Please see also our response to point #5.**

2. It is not made explicitly clear that when the number of tiles is greater than the number of disturbance events in a gridcell, the results for the gridcell will not improve (or change at all) with allowing more tiles (because they won't be used). I assume the authors understand this, but the closest that the manuscript says to this is lines 426-433, although they don't say this clearly. Instead, the text talks about a "roughly exponential decline" and mentions "saturation". This text is very wordy (it is hard to read) and misses the most important point: when you increase the number of dynamics tiles to 7, the disturbances in most gridcells are perfectly (or near perfectly) resolved and the results agree with the 32 tile run (for most variables). This should be mentioned and its impact on the conclusions better discussed.

**This is a good point, we have revised the text within this section to highlight that when 7 – 12 tiles are included, the simulations roughly approach the 32 tiles run. We have clarified that the model with 32 tiles can capture every discreet disturbance event from 1985 – 2017, but that capturing every disturbance event starting in 1700 would be computationally intractable. Finally, we discuss the exponential declines in msd as evidence that increasing the number of tiles rapidly minimizes discretization error and approaches a convergent solution as the model can represent more patches of vegetation in different stages of recovery. The revised text near line 451 reads "The change in the msd (Figures 4a-h) as the maximum number of available tiles for the run increases from 1 to 32 exhibits a roughly exponential decline for surface energy balance (HFSS, HFLS) and C cycle-related variables (cLand, GPP, ER, LAI). The msd is near zero at 7 – 12 tiles. The 32-tile simulation captures**

**all the discrete disturbance events from 1985 - 2017 across most of the model domain (Figure 2). However, a simulation that resolves all the disturbance events between 1740 and 2017 as tiles would require far more than 32 tiles, in many forested areas, and be computationally intractable. However, we infer from the exponential (e.g. rather than linear) decreasing rate of change in figures 4a-f that our reference 32-tile simulation has minimal discretization error and converges on the results of that computationally intractable simulation (Torres-Rojas et al., 2022; Nabel et al., 2020; Nocedal & Wright 2006). The difference in msd between the 32-tile tile simulation and that computational intractable simulation would likely be vanishingly small, similar to the difference between the 25-tile and 32-tile simulations (Nabel et al., 2020; Fisher et al., 2018; Ellner and Guckenheimer, 2011; Gelman and Hill, 2006). These roughly exponential declines in msd reflect the model's ability to discretize patches of vegetation in different stages of recovery using greater numbers of tiles. This is reflected in how the statistical distributions of aboveground tree biomass in forested grid cells change as more tiles are utilized in the simulation (Figure 5a)."**

Relatedly, I need to flag the statement in the Discussion (line 528): "Our results suggest that representing a relatively small number of heterogeneous tiles (e.g. < 12) may yield undesirable biases when compared to simulations using a larger number of tiles (32-tile; Figures 4a-h, 6a)". Based on Fig 4, the 7-tile simulation pretty much nails the 32-patch run and so doesn't yield "undesirable biases". The exception is the fire CO2 emissions but these emissions are very insensitive to the tiling (Fig 6h). Also, Fig 6a) only shows comparisons of 1 vs 32 tiles, values like 7 and 12 are not shown, it cannot support the statement. Based on the two points above, I think the discussion of optimal number of tiles needs to be reconsidered.

**Thank you for pointing this out there is a typographical error in the figure reference, and it should read 5a, we have corrected this error. We have also modified this text to provide further clarity regarding our determination of the optimal number of tiles and now reference a range from 7 – 12 tiles. Seven tiles corresponds to the start of the range at which most of the plots in figure 4a-h asymptote, whereas twelve tiles is well within the asymptote. This also falls in a range of reasonable computational costs. We have modified the text near line 568 to read "Our results suggest that representing a relatively small number of heterogeneous tiles may yield undesirable biases when compared to simulations using a larger number of tiles (Figures 4a-h, 5a) (Yue et al., 2018a; Shevliakova et al., 2009; Stocker et al., 2014; Yang et al., 2010). For a tile-based LSM to represent these subgrid impacts the simulation needs to be sufficiently complex and judiciously implemented and tested. In the case of CLASSIC, we find that 7 - 12 tiles optimally balances detailed representation and computational costs. An *rht* of 0.16 and a *tpp* of 4 increases computation efficiency with little impact on the level of detail represented."**

3. The manuscript is rather long, but despite all the words, the wording is not always clear and can be difficult to follow. Here is an example starting at line 344:

"We utilized aspatial records of the total harvested and burned area within Canada to bias-correct inferred disturbance from 1920 - 1984 (Skakun et al., 2021; World Resources Institute, 2000). Before 1920, we utilized aspatial records of total disturbed area derived from 1920 stand age, with harvest held constant (0.3 Mha yr -1 ) (Chen et al., 2000; Kurz et al., 1995). First, for years

in which inferred burned or harvested area (D inferred for i years, l grid cells; m 2 ) exceeded the aspatial records (d aspatial for i years; m 2 ) we correct the positive biases."

Problem I encountered when reading this:

a) The order of the first two sentences should be swapped because they don't match the conceptual temporal ordering (i.e. should be pre-1920 and then 1920-1984). In fact, both sentences could be combined into something much more succinct since they repeat a lot of words and deal very much with the same topic.

**This section has been heavily revised see our responses to points b-e and the overall prompt below.**

b) It is not immediately clear if the bias correction was also applied to the pre-1920 data or not despite the data getting its own sentence. Likely not, but the sentence describing this data comes in between the first mention of the bias-correction and the description of it, so the implication is that it was in fact applied to pre-1920 data…? I don't know what to make of it.

**We have clarified that we "utilize the aspatial records to bias-correct the 1740 – 1984 disturbance that has been inferred from stand age" (line 352).**

c) Getting into the description (third sentence), what is the "inferred" burned area or harvested area?

**We have clarified that we are referring to "disturbance inferred from stand age" (line 357).**

d) Why are positive and negative biases being corrected differently? No explanation is given.

**We have added an in-depth explanation of the logic behind these different methods near line 356. "We utilize bias correction that retains the spatial patterns of pre-1984 disturbance inferred from stand age while correcting positive and negative biases to match the aspatial records. This necessitated two distinct bias correction methods. For years with positive biases, the positive bias indicates that there is sufficient disturbance inferred from stand age. In this case, a uniform bias correction factor can be used to scale down disturbance. Years with negative biases, however, do not contain sufficient disturbance as inferred from stand age, Here residual disturbed area from nearby years needs to be added to the year under consideration to match the aspatial record level of disturbance while preserving the spatial patterns derived from the stand age. Because the uncertainty of stand age estimates increases further into the past, the negative bias correction is carried out by starting in 1984 and looping backward annually in time until 1740 (Maltman et al., 2023)."**

e) In general, what follows this text is a (confusing) technical explanation of the bias correction procedure. At no point were the goals of the bias correction mentioned,

potential pitfalls, the rationale for the choices made, etc. And what is this "loop backward in time"? This is a completely new idea to me, why was this done?

**We have revised this section of the methods to increase clarity and clarify the goals and methods of the bias correction as well as the reasoning behind the different bias correction methods. See the text above as well as the introduction to the paragraph beginning near line 351 which states "However, pre-1984 disturbance that has been inferred from stand age does not align with available aspatial records of total harvested and burned area within Canada. Therefore, we utilize the aspatial records to bias-correct the 1740 – 1984 disturbance that has been inferred from stand age to ensure the total values match the available historical records."**

As mentioned above, another example can be found in lines 426-432. Many words are used to basically say "When there are more tiles than disturbances, the results won't change if you add more tiles.

**We have revised this section for clarity. We more formally discuss how the exponential declines in msd evidence that increasing the number of tiles rapidly minimizes discretization error and approaches a convergent solution as the model can represent more patches of vegetation in different stages of recovery. (See the edits in our response to point number 2)**

These are just two particular examples. The text is afflicted throughout with poor and non-logical flow, awkward wording, and inappropriate levels of detail.

Because of the above, large parts of the manuscript should be critically reviewed and rewritten.

Some general suggestions:

- Use more subheadings. This will force the writer to focus more precisely on what that text is supposed to say and will help the reader to understand exactly what they should be taking from the text.
- Describe the only methods briefly in the main text but move the details to the appendix. This will improve the flow significantly.
- One specific point, please include more high-level details of the "normalized response metric". How should the values be interpreted its their scale? And maybe give one sentence summarising the construction of the variable before jumping in the technical details.

**We have revised the text to utilize more subheadings to distinguish important sections within the methods and revised details within the methods to provide a more high-level overview of what we did (see lines 310 - 439). Finally, we have added more high-level details regarding the normalized response metric when it's introduced in the methods. These details added near line 426 read: "The normalized response metric is a unitless summary statistic. Its strength is that a wide range of variables with different units can be visualized on the same axis to make relative comparisons of their simulated responses to disturbance and tiling."**

4. The overall quality of the language is not too bad in terms of grammar and style, but it needs tightening up. Examples (non-exhaustive):

**Please see our response to these individual points below. We have also made further revisions to the text for clarity throughout.**

- Line 125 - "We use a domain, which encompasses all of Canada south of 76°N as our study area for demonstration." Technically correct but very awkward.

  **This sentence near line 124 has been streamlined to read "We use all of Canada south of 76°N as our simulation study area"**

- Line 191-195 – This single sentence is huge (fully four lines with not a single comma) and completely cryptic. Reformulate.

  **This sentence near line 193 has been split, edited, and streamlined to read "Because the maximum number of tiles is fixed, the model must manage the number of tiles being actively simulated. The model ensures that up to two inactive tiles are available to simulate disturbance each year (i.e. one for fire and one for harvest; see section 2.3.4)."**

- Line 236-238 – Not really a proper sentence.

  **This sentence near line 235 has been revised to read "By default, the model selects the two tiles with the most similar vegetation heights and joins them."**

- Line 417 – "Alternately" is not what the authors mean. I think "In contrast" or "Contrastingly".

  **"Alternately" has been replaced by "In contrast" in line 442**

- Line 466 – Why are there citations behind an assertion about the paper's own results?

  **The citations has been removed.**

- Line 524 – Use of the term "biases". This is a matter of taste, but I dislike using "bias" when referring to model-to-model comparisons and prefer to reserve it for model-to-data comparisons. The authors may want to consider a different term.

  **We agree that bias is not the correct term in the context of model-to-model comparisons and should be reserved for broader discussions of the biases that may stem from the absence of dynamic tiling or dynamic tiling setups. Therefore, we've removed it in this context around lines 20 and 116.**

5. There is very little comparison of the model improvements to data (only Fig 5b), but a large amount of model-to-model comparisons. The stated logic is that the 32-tile simulation with disturbance enabled is the best possible simulation, and therefore the standard to which the other simulations should be compared. But is this 32-tile, disturbance-enabled simulation actually any better when compared to data than a single-tiled, non-disturbed simulation? Logically it might be better, but highly generalized but complex process-based models (such as LSMs) might not respond the way one expects. Some further model-to-data comparisons should be considered to

make sure that changes are not actually making the model skill worse due to pre-existing cancellation of errors or some bias.

**Herein we focus on model-to-model comparisons to assess the impacts of more detailed representation using more tiles on discretization error and gain insight into the model configuration and and role of these processes within CLASSIC. This is a step towards utilizing this model for C cycling assessment work. Our exclusive use of model-to-model comparisons allows us to focus on these questions and the biases induced by subgrid-scale heterogeneity like other work that compares idealized simulations of varying complexity to address ecological/numerical questions (Torres-Rojas et al., 2022; Moorcroft et al., 2001). It also allows us to leverage counterfactual simulations, with disturbance absent to develop ecological insights. In this context, model-to-data comparisons would: 1) require data sets with an extremely high spatial resolution (<1km$^2$) and information content (e.g. spatial patterns that are not a product of interpolation alone), which are likely not available (see: Curasi et al., 2022 for an overview of available data set resolutions). 2) The pre-existing biases in the model, uncertainty in the data itself, or uncertainty in the model drivers might obscure the role of subgrid-scale heterogeneity. This could lead us to select optimal tiling parameters that eliminate these biases, rather than minimizing discretization error as we have done here.**

**We have revised the text near line 406 to better explain this approach "We carry out model-on-model comparisons for a selection of variables and model configurations for the satellite era portions of our simulations (1985 - 2017) to select the model setup that optimally balances detailed process representation and model run time (Table 3). This model-on-model approach has the benefit of canceling out any pre-existing biases in the model and focuses our evaluation on the impacts of subgrid-scale heterogeneity and discretization error alone (similar to Torres-Rojas et al., 2022; Moorcroft et al., 2021). We also use these evaluations to demonstrate the relative impact of representing subgrid-scale heterogeneity within our modeling framework." We have also added discussion of this near line 563 "These results are strengthened by our model-on-model approach which acts to cancel out pre-existing biases to demonstrate the impacts of subgrid-scale heterogeneity, and discretization error alone (Torres-Rojas et al., 2022; Curasi et al., 2022; Melton et al., 2017; Melton & Arora 2014; Moorcroft et al., 2001).".**

Relatedly, the explanation of the pre-existing "mosiac" tiling feature is not really clear (line 171). It is not specified how the tiles differ from one another – one must assume it varies across the cited studies. Giving more details about these studies and briefly summarising how mosaic tiling improved the model compared to the default "composite" approach would help justify the near-exclusive use of model-to-model comparisons in the present study.

**The mosaic tiling feature refers to simulations utilizing more than one tile. As a result, it is highly flexible and has been used to address an array of different questions. We have included some additional detail regarding these studies as well as some contrast between the approaches utilized in these studies and our approach near line 176. "CLASSICs tiling capability has been used in the past to investigate the impacts of subgrid-scale heterogeneity in soil texture by breaking grid cells with heterogenous soil textures into tiles (Melton et al., 2017). As well as vegetation cover (Melton and Arora, 2014; Li and Arora, 2012), and competition between plant functional types (Shrestha et al., 2016) by breaking**

**grid cells with heterogenous vegetation cover into tiles. These approaches result in regional differences in fluxes of up to 30%. We adapt the mosaic representation to dynamically create disturbance history tiles and represent the subgrid-scale heterogeneity resulting from disturbance (i.e. represent a complete harvest of an area corresponding to 50% of the grid cell as a 100% reduction of the vegetation biomass in a newly created subgrid tile that covers 50% of the grid cell; Figure 1). In our approach, the tiles serve to represent vegetation that is in different stages of recovery. Thus the soil textures and vegetation fractional cover are the same for all tiles within a given grid cell."**

6. The authors made a point of describing the max of 66 disturbance events (due to the 33 years of the evaluation period) and relating that to the number of tiles. Simultaneously they spend a lot of effect deriving historical disturbance scenarios and also clearly state that disturbance is applied throughout. But they don't make it clear how the disturbance is applied in the pre-evaluation period (since the can't be contributing to the maximum 66 events). After carefully re-reading, it seems likely that the composite representation was used in the earlier period but I don't think this is stated anywhere. If it was, will this transition from composite to mosaic change affect the results, particularly in the early phase of the evaluation period? Please make it clear and discuss.

**Thank you for pointing out this unclear point. We have added a statement near line 401 to clarify that disturbance is applied the same way between the pre-evaluation and evaluation period: "The 14 transient simulations utilize their individual unique, land surface representation (i.e. composite or mosaic), maximum number of available tiles, *rht*, and *tpp* for the entire 1700 – 2017 run (Table 3)." We did this to avoid a sharp transition as you mentioned above. The reasoning behind using 32 tiles is two-fold. First, it is informed by the maximum number of disturbance events within the evaluation period with the intention that the model is capable of discreetly representing all these events. Second, it is informed by computational limitations as 32 tiles is at or near the upper limit of what we can run reliably (see line 474). Also see our response to points number 2, and 5 above for our comments and revisions related to the interpretation of the model-on-model comparisons.**

7. There is no mention of other disturbance agents such as biotic agents (importantly bark beetle), drought, wind throw, and landslips. It is fine that they weren't included (not everything can be modeled at once) but some discussion is needed. Why were certain disturbances chosen and not others? Could this approach apply to other disturbance agents? What do the authors believe the consequences of leaving these out could be?

**This is likewise a valuable point. We chose widespread disturbances, known to be important for Canadian and global C cycling, stand-replacing, and that have spatially explicit time-series data from which a forcing can be created. We have added additional discussion near line 551 that discusses the types of disturbance that could be included in the future, practical considerations therein, and further interpretation of our results through this lens. "Other disturbances including insect damage, wind damage, and landslides could likewise be represented using dynamic tiling. Insects in particular are an important disturbance agent in Canada that have more widespread impacts than fire and harvest, but greater variation in severity (Kurz et al., 2008; Chen et al., 2000). Representing these disturbance events requires consistent spatially explicit time series of the forcings, which are not widely available at present (Pongratz et al., 2018; Erb et al., 2017). This would also**

**require careful consideration of the impacts of the disturbance in question. We can infer from our results that low-severity non-stand replacing disturbances may not require a tiled representation."**

*Reviewer #2*

Summary

The authors present a subgrid tiling method for a land surface model to improve simulations that include fire and harvest disturbances. Each grid cell is divided into tiles based on two potential disturbances per year. Each tile has a representative set of pfts (including veg height) and a time since disturbance. The authors show that multiple tiles (as opposed to a 'single' tile) can have a dramatic impact on biogeochemical outputs, and that increasing the maximum number of tiles eventually reaches an asymptote with respect to changes in outputs. The authors further explore dynamic tiling parameters to find an optimal configuration. They conclude that this is a viable approach for land surface models to reasonably capture more details associated with subgrid vegetation disturbance processes.

Overall response

This is a good example of model advancement that increases detail and complexity to achieve greater accuracy without requiring major restructuring of input data. It also demonstrates how much of a difference it can make to try to more accurately represent vegetation change at a subgrid level. It does require some clarification, I am not convinced by the choice of optimal setup, and I it is unclear whether this method actually increases the accuracy of the model. My main concerns are outlined here, with more detailed comments following.

**Thank you for reviewing our article and providing comments. We appreciate your review's recognition of the value of our work and the opportunity to further improve the text. Below are our responses to your comments and associated edits to the text.**

1) Some of the text is unclear, particularly in the methods section. See details below.

**We have revised a large portion of the methods and main text for clarity. See our table of detailed responses below.**

2) The optimal configuration is selected simply for computational efficiency, rather than taking into account the potential model response. But the response analysis shows that the responses can be quite different, while the computational efficiency appears nearly the same for all 12-tile configurations. If the reference configuration is truly believed to be a more accurate representation of the processes, then it should factor more strongly into this decision. In particular, the two disturbance outputs have a very poor response with the chosen optimal configuration, in relation to the reference. One challenge here is that there is no accuracy or skill assessment, so selecting an optimal configuration is lacking the dimension of model accuracy (see next point). Another is the units of the computational efficiency: the reader cannot tell whether a one second difference per cell actually matters. Doe this difference mean the model takes either 15 or 18 hours to run 30 model years, or 5 or 18 hours to run 30 model years? If it is the former, then you want the more accurate configuration. If it is the latter then you have to consider resource tradeoffs.

We strive to balance computational complexity and detailed representation in this work. The computational cost of running CLASSIC with tiles is considerable and factors into our determination of the optimal configuration as do the analyses shown in figure 4, 6, S2, and S3. The CLASSIC framework is intended to be run in a wide range of configurations from a serial configuration where only a single point location is simulated to parallel runs at the global scale across multiple nodes of a computing cluster. To generalize the computational cost estimation, we normalize the run time so the information provided is as broadly applicable as possible. However, to provide some numbers here, just counting the modern portion of the run used in our primary analyses, CLASSIC takes around 318 hours of CPU time (i.e. the sum of time utilized by all cores across multiple machines) for a single tile run. That cost increases as the number of tiles increases, but not linearly. The 32-tile run takes about 14 times as long as a single-tile run, the 12-tile run takes about 3 times as long, and the optimized run takes 2 times that of a single-tile run. It is thus important to limit the number of tiles to ensure computational costs remain reasonable. Concerning accuracy most of the plots in figure 4a-h asymptote around 7 tiles. Two variables asymptote closer to 12 tiles: fFire and fDeforestedTotal, however, as we lay out the results these plots are influenced by the relatively low magnitude of difference between the simulations, which is more clearly visible in figure 6 (a, h, i) than in figure 4. The differences shown in figure S2 are all relatively small for these variables as shown in figure 6 and S3. Finally, our comparison to National Forest Inventory aboveground biomass data in figure 5 provides clear insight into the role of our model developments and these parameters in improving the representation of recently disturbed forests. That is when looking at the portion of the aboveground biomass histogram between 0 and ~5 gC m$^{-2}$ representing disturbed areas to a sufficient degree of detail enhances the correspondence between the modeled and observed statistical distribution.

We have modified the conclusion text to provide further clarity regarding our criteria for determining the optimal number of tiles and optimization parameters. It reads "In the case of CLASSIC, we find that 7 – 12 tiles optimally balances detailed representation and computational costs. An *rht* of 0.16 and a *tpp* of 4 increases computation efficiency with little impact on the level of detail represented.". We have also added information about the model run time in the results to provide further background for the reader "The run time for the satellite era simulation (1985 - 2017) with 1-tile is ~318 CPU hours (i.e. the sum of time utilized by all cores across multiple machines; Xeon Platinum 8380). Compared to the 1-tile run, the 32-tile, 12-tile, and optimized run consume 14 times, 3 times, and 2 times as many CPU hours, respectively." (line 473)

3) Does this structural advancement improve model accuracy? The assumption is that by representing finer resolution disturbance the accuracy of the simulation should improve. But the one comparison with above ground tree biomass does not indicate any model improvement with this structural change, but it does take more computational resources. So how do you justify the increased complexity? Key outputs are clearly affected by this approach, but do you want to use this approach if it reduces model skill? This may or may not be required in the context of GMD, but I suggest running your outputs through some sort of benchmark or skill assessment to show that this approach is a worthwhile advancement.

In our study, to assess the impacts of a more detailed representation of sub-grid heterogeneity using more tiles, we focus on model-to-model comparisons of discretization error and use the comparisons to gain insight into the model configuration and role of these processes within CLASSIC. Our exclusive use of model-to-model comparisons allows us to focus on these questions and the biases induced by different methods of accounting for subgrid-scale heterogeneity in a manner similar to other work that compares idealized simulations of varying complexity to address ecological/numerical questions (e.g. Torres-Rojas et al., 2022; Moorcroft et al., 2001). It also allows us to leverage counterfactual simulations, with disturbance absent, to gain ecological insights. We chose to make the comparison to National Forest Inventory aboveground biomass data in figure 5 because it provides clear ecological insight and illustrates how the model improves the representation of recently disturbed forests compared to the observation-based data when looking at aboveground biomasses between the range of 0 and ~5 gC m$^{-2}$. Further, model-to-observation-based data comparisons would require data sets with an extremely high spatial resolution (<1km$^2$) and information content (e.g. spatial patterns that are not a product of interpolation alone), which are likely not available (see: Curasi et al., 2022 for an overview of available data set resolutions). Moreover, the pre-existing biases in the model, uncertainty in the observation-based data itself, or uncertainty in the model drivers might obscure the role of subgrid-scale heterogeneity. This could lead us to select optimal tiling parameters that best reduce these biases (which may not lead to a more truly 'realistic' simulation), rather than minimizing discretization error as we have done here. We have revised the text near line 406 to better explain this approach "We carry out model-on-model comparisons for a selection of variables and model configurations for the satellite era portions of our simulations (1985 - 2017) to select the model setup that optimally balances detailed process representation and model run time (Table 3). This model-on-model approach has the benefit of canceling out any pre-existing biases in the model and focuses our evaluation on the impacts of subgrid-scale heterogeneity and discretization error alone (similar to Torres-Rojas et al., 2022; Moorcroft et al., 2021). We also use these evaluations to demonstrate the relative impact of representing subgrid-scale heterogeneity within our modeling framework." We have also added discussion of this near line 563 "These results are strengthened by our model-on-model approach which acts to cancel out the influence of pre-existing biases to demonstrate the impacts of subgrid-scale heterogeneity, and discretization error alone (Torres-Rojas et al., 2022; Curasi et al., 2022; Melton et al., 2017; Melton & Arora 2014; Moorcroft et al., 2001).".

**Specific suggestions/comments:**

| Specific suggestions/comments: | Response: |
|---|---|
| Abstract line 30: But you don't show what the model biases are or whether they are reduced. So is there improvement? | Please see our response to point number three above. However, we agree that bias is not the correct term in the context of model-to-model comparisons and should be reserved for broader discussions of the |

| | |
|---|---|
| | **biases that may stem from the absence of dynamic tiling or dynamic tiling setups. Therefore, we've removed the word 'bias' in this context around lines 20 and 115. For example, line 20 now reads "We then demonstrate the impacts of subgrid-scale heterogeneity relative to standard average individual-based representations of disturbance and explore the resultant differences between the simulations."** |
| Introduction line 120: What processes are you referring to? all of the ones mentioned in this paragraph (ranging from disturbance to energy flux to model algorithms)? additional ones previously mentioned (e.g, vegetation productivity). The most recent processes mentioned are tile creation and merging. | **This sentence has been revised for clarity and now reads "This investigation provides insight into the model configuration and role of fire, harvest and tiling these processes within CLASSIC" (line 119).** |
| Methods line 125: "Our study domain encompasses all of Canada south of 76N" | **This sentence has been revised to read "We use all of Canada south of 76°N as our simulation study area" (line 124).** |
| line 130: "In Canada, annual, contiguous timber harvest events remove 98+-…" | **This sentence has been revised to read "In Canada, over the course of a year, each contiguous timber harvest event clears on average $98 \pm 115$ ha." (line 128).** |
| line 132: see line 130 | **This sentence has been revised similarly to the above (line 130).** |
| line 135: How does harvest account for only 0.2% of stand replacing disturbance if 52% of the forest is managed? | **"Managed forest" is widely used in related literature including Stinson et al., 2011 and 2019. It does not refer to locations that are being actively harvested but rather the total area of land managed for potential timber harvest, under protection from disturbance and conservation areas.** |
| line 141: Be clear that CLASSIC couples CLASS and CTEM; according to the next paragraph, CLASSIC isn't merely based on them. | **The text has been edited to clarify that "CLASSIC is an open-source community model that couples the Canadian Land Surface Scheme (CLASS) (Verseghy, 2000,** |

| | 2017; Verseghy et al., 1993; Verseghy, 2007) and the Canadian Terrestrial Ecosystem Model (CTEM) (Melton and Arora, 2016; Arora, 2003)." |
|---|---|
| line 156: does "canopy-covered ground" mean that it does canopy energy exchange, or is this done by CTEM? | This text has been edited to clarify that "CLASS simulates ground and canopy energy exchange from four possible subareas: bare ground, snow-covered bare ground, canopy-covered ground, and snow-covered canopy, on a thirty-minute time step." |
| lines 175-182: several other subgrid papers exist. here are a couple of examples.

subgrid and surface energy balance: Hao et al 2022. Impacts of Sub-Grid Topographic Representations on Surface Energy Balance and Boundary Conditions in the E3SM Land Model: A Case Study in Sierra Nevada. james, 14(4):e2021MS002862. https://doi.org/10.1029/2021MS002862

subgrid and water, fluxes, energy balance: Singh et al 2015. Toward hyper-resolution land-surface modeling: The effects of fine-scale topography and soil texture on CLM4.0 simulations over the Southwestern U.S. water resources research 51(4):2648-2667. https://doi.org/10.1002/2014WR015686 | It was unclear in the original text, that we are reviewing past uses of CLASSICs tiling capability. The text has been revised for clarity and to provide greater detail. "CLASSICs tiling capability has been used in the past to investigate the impacts of subgrid-scale heterogeneity in soil texture by breaking grid cells with heterogeneous soil textures into tiles (Melton et al., 2017). As well as vegetation cover (Melton and Arora, 2014; Li and Arora, 2012), and competition between plant functional types (Shrestha et al., 2016) by breaking grid cells with heterogeneous vegetation cover into tiles. These approaches result in regional differences in fluxes of up to 30%." (line 179). |
| lines 206-207: While it is technically necessary for the new tile to not exceed the available space, this limit does not make sense in this context because it would require all of the candidate tiles to be merged to reach this limit. There is a semantic challenge here where "splitting" multiple tiles also requires merging the split-off areas. You may consider tile "creation" and "joining." | This section and equation 1 have been revised to clarify that "the fractional area occupied by the single new tile must be less than the sum of the vector of fractional areas of the candidate tiles" as opposed to less than or equal to (line 207).

We agree that there is a semantic challenge here. We chose to avoid the use of "creation" because it may imply that the operation doesn't conserve mass, energy, etc. We have removed it in locations where it was previously used in reference to tiling. In practice, splitting and joining of tiles are |

| | |
|---|---|
| | **accomplished by applying similar calculations to the model state variables (i.e. eqn 3 and 4). The key difference is the way tile fractional area is considered. Therefore we use "splitting" to highlight that the operation leaves some fraction of the candidate tiles behind and "joining" to highlight an operation that utilizes the entire fractional area.** |
| line 233: eq 4 appears to be equal to a unitless 1. I think you need to remove the t term from the denominator. | **Thank you for pointing this out. The typographical error in equation 4 has been corrected by removing the t term in the denominator.** |
| lines 234-253: The description and variables do not match the equations, which makes this section confusing. | **We have revised equation 5 so the PFT index is used more consistently.**

**We have also modified the description of the nested iterators n1 and n2. That is "The model uses the vector of tile average vegetation heights (ħ, of length $n$ for a total number of tiles; m) and calculates the absolute difference between all possible combinations of the elements therein (i.e. using the nested iterators $n1$ and $n2$). The resulting absolute difference matrix of tile average vegetation heights ($\Delta\bar{H}$ a $n1$ total number of tiles by $n2$ total number of tiles matrix; m) is used to judge the similarity between tiles."**

**Finally, for clarity, we have also revised equation 6 to show the comparisons between the matrix $\Delta\bar{H}$ and the single value rht ∗ max (ħ) which would result in a boolean matrix rather than showing the iterators n1 and n2 again so this equation follows the description more closely.** |
| lines 234-253: It also is not clear that there is only one parameter for allowing merging, and then a second one for preventing merging. When and how are the rht and tpp set? is tpp a minimum number of total tiles to keep? Does | **We have revised the tile management section for clarity. We now describe the default case explicitly in the first paragraph "In the default case the two tiles with the minimum $\Delta\bar{H}$ are joined when the** |

| | |
|---|---|
| tpp simply retain the shortest veg tiles in order to meet this minimum number? is rht the threshold, or is there additional calculation required to get the threshold (and how is it calculated)? what happens if rht and/or tpp are not set? | **maximum number of dynamic tiles is reached." (line 242). We have also added further detail regarding *rht* to tie it into the logic of equation 6 namely "An optional relative height threshold (*rht*; unitless) allows for tiles to be pre-emptively joined at a yearly time step before reaching the maximum number of dynamic tiles. The *rht* can be conceptually thought of as breaking the tiles into equally spaced bins organized by vegetation height.". We have also clarified the role of the *tpp* namely "When the *rht* parameter is used, the optional tile preservation parameter (*tpp*; number of tiles) prevents tiles with the shortest average vegetation height from being merged. That is the model, starting with the tile with the shortest average vegetation height, retains that number of tiles, *tpp*. This means the tiling scheme will carry out pre-emptive joins based upon *rht* while preserving young, recently disturbed tiles and explicitly representing early successional differences in fluxes (Bellassen et al., 2010; Zaehle et al., 2006; Nabel et al., 2020)." (line 254).** |
| lines 296-392: you may want to reiterate that these fractions are specific to Canadian forest harvest and processing. | **These fractions were derived by Arora and Boer 2010 and are suitable for use globally. If more detailed regional information becomes available in the future these fractions could be modified. We have revised the text to clarify the source of these fractions "In either case, the harvested aboveground biomass (i.e. both non-structural and structural stem and leaf C) is split into three streams using fractions developed by Arora and Boer 2010." (line 303)** |
| lines 313-320: if this is static land cover, what year(s) is it based on? | **The methods now note that "This land cover corresponds to the year 2010" in line 323.** |
| line 318: prescribed land cover can vary over time; this is static land cover | **Line 328 has been corrected to read "(i.e. static land cover as opposed to dynamic or** |

| | prescribed land cover changes)" |
|---|---|
| lines 358-366: this is unclear and confusing. figure 3 helps somewhat. | We have revised this section substantially to include additional background and details regarding these methods. First, we have added background text to provide an overview which states "We utilize bias correction that retains the spatial patterns of pre-1984 disturbance inferred from stand age while correcting positive and negative biases to match the aspatial records. This necessitated two distinct bias correction methods. For years with positive biases, the positive bias indicates that there is sufficient disturbance inferred from stand age. In this case, a uniform bias correction factor can be used to scale down disturbance. Years with negative biases, however, do not contain sufficient disturbance as inferred from stand age. Here residual disturbed area from nearby years needs to be added to the year under consideration to match the aspatial records level of disturbance while preserving the spatial patterns derived from stand age. Because the uncertainty of stand age estimates increases further into the past, the negative bias correction is carried out starting in 1984 and looping backward annually in time until 1740 (Maltman et al., 2023)." Next, we have revised the description of the actual procedure for clarity. "Second, for years in which burned or harvested area inferred from stand age falls below that indicated in the aspatial records, we correct the negative biases by adding in the residuals from nearby years (Figure 3). We loop backward in time from 1984 to 1740 and accumulate residuals ($r_{moving}$ for $l$ grid cells; $m^2$) extending as far back in time as needed to exceed the aspatial record for the year under consideration ($d_{aspatial}$, $i$). We calculate an aspatial bias-correction factor ($f$; unitless) and use it to apply a fraction of $r_{moving}$, to the inferred disturbance time series and |

| | |
|---|---|
| | **subtract the residuals used from r$_{moving}$. When the spatially explicit residuals are exhausted (~1920 for fire only) they are replenished using the entire gridded remotely sensed and stand age inferred disturbance record. This procedure continues until all the negative biases have been corrected between 1984 and 1740 yielding the final spatially explicit time series (D$_{final}$ for i years, l grid cells; m$^2$).” (lines 331 - 387).** |
| lines 370-375: this information should probably be up where rht and tpp are introduced. it should help clarify what rht and tpp are, what they mean, and how they are used. here rht and tpp sound very different from how they are introduced. also, how is preemptive merging different from regular merging. the previous section describes only preemptive merging. | **We have revised the tile management section for clarity. We now describe the default case explicitly in the first paragraph and provide greater detail regarding *rht* and *tpp*. See our response for lines 234-253 above.** |
| line 382: this is related to lines 322-366. i appreciate that you do your best to develop fire and harvest drivers for the entire simulation, but you end up with three very different disturbance regimes for each, and dramatic singularities at the transitions between regimes. have you looked at how these different regimes and transitions affect your simulations? very different things are happening in each regime, and the cumulative effects are going to give you a unique state at each transition. how would your post-1985 outputs look if you simply fixed historical fire and harvest to the first observed level? or even repeated the observed pattern? the singularities can cause dramatic shifts in your model outputs due to such large and likely unrealistic changes in disturbance regime. these effects can propagate over time in your simulations and generate large uncertainties in your results. | **We chose to reconstruct historical disturbance using the best data sources available and explicitly highlight the methodology and results therein as we feel this brings a more realistic test case for the tiling framework. The alternative would have been to do model-on-model experiments using only idealized scenarios (i.e. the average of the observed applied uniformly across space). Due to our adopted approach, we rely on stand age, and aspatial historical records and treat those historical records as authoritative and accurate. We have work ongoing to unify a wider range of disparate spatial and aspatial historical records and reconstruct the trajectory of fire and harvest over time. This will be important for using this model to simulate the historical trajectory of the Canadian carbon cycle.**

**The questions addressed in the manuscript and the types of experiments carried out** |

| | further mitigate any uncertainties resulting from historical disturbance. We demonstrate the impacts and tradeoffs involved in representing disturbance and sub-grid scale heterogeneity with tiles. We use model-on-model evaluations where all the runs have the same fire and harvest trajectory and focus on the period from 1985 to 2017 when observations drive the model. In figure 6 rather than evaluate the impact of tiling in absolute terms we evaluate it relative to the impact of disturbance. Overall, the analysis was done in such a way that any modifications to the pre-1985 fire and harvest would impact all the simulations therein and therefore yield results that are robust to the drivers. |
|---|---|
| lines 394-410: This is confusing. the terms are not consistent and it appears you have redundant terms. at the beginning the 32 tile run is the reference and at the end it is a target. What is a target run (it sounds like one of the 14 simulations you want to evaluate)? what are j model runs? there isn't any indication that you run the model multiple times for each target simulation. | We have removed the term target from equations 11 and 12. Equation 11 details all 14 model runs (j) being compared to the 32-tile run. Each scenario in Table 3 consists of a single model run. We have also added text specifying that the j model runs refer to those "detailed in table 3". |
| You don't have a 32-tile run, unless it is mislabelled in table 3. or it needs to be added to table 3 as the reference run. | Thank you for pointing this out. We have corrected the typographical error in table 3 to reflect the 32-tile run. |
| equation 11 has n years and i years, and equation 12 has an m index that is not defined. also, equation 11 isn't mean square error because your reference is arbitrary (it is a mean square difference); you are not comparing against observations to determine the accuracy of the model. | We have revised equations 11 and 12 for clarity including replacing n years with its value "33", removed the "m" index which was a typographical error, and we now use "mean squared deviation" in place of "mean squared error". |

| | |
|---|---|
| lines 420-421: this is somewhat due to your input data in that you can have at most one fire event and one harvest event per year. | **This upper limit is important in the context of what follows and is a product of the model structure. We've revised this sentence to clarify that "Our aspatial tiling scheme operates on an annual timestep and therefore the maximum number of possible events in the 1985 - 2017 drivers is 66 (i.e. a harvest and fire each year for 33 years; Figure 2e)." (line 447)** |
| lines 440-441: not really. the following two sentences still make sense, though, with respect to the overall response. | **This text reflects that in figure 4 a-f msd reaches its asymptote at 7 tiles with a sharp decline from 1 to 3 and then 7. In the case of 4 g-h the decline from 1 to 3 and then 7 tiles is "less sharp". The remaining text has been revised to better reflect the plot and state that the msd then "approaches zero at 12 tiles". (line 466)** |
| line 445: incomplete sentence | **This sentence has been revised for clarity and reads "There is a gradual increase in msd for fFire and fDeforestedTotal as *rht* and *tpp* increase (Figures S2g-h)." (line 486)** |
| lines 453-466: what about fire emissions and deforested c (figure 4)? these two variables are quite far off from the increasing tile trajectory. these are also directly related to your primary goal of simulating fire and harvest disturbance. it would make more sense to optimize for these variable outputs in conjunction with the others, unless you don't believe that the 32-tile simulation is accurate. tpp should be 6 and rht should be 0.04. especially since the run time is similar across all combinations of these. or maybe these two shouldn't be set all because the the 12-tile with these unset has a similar runtime also (but a couple variables have higher difference from 32-tile). | **We have modified the conclusion text to provide further clarity regarding our criteria for determining the optimal number of tiles and optimization parameters. It reads "In the case of CLASSIC, we find that 7 – 12 tiles optimally balances detailed representation and computational costs. An *rht* of 0.16 and a *tpp* of 4 increases computation efficiency with little impact on the level of detail represented."** **The computational cost of running CLASSIC with tiles is considerable and factors into our determination of the optimal configuration. Seven tiles correspond to the start of the range at which most of the plots in figure 4a-h asymptote, with the vast majority asymptoting at twelve. We highlight in the text that the patterns in the fFire and** |

| | |
|---|---|
| | **fDeforestedTotal are "influenced by the relatively low magnitude of the differences between the simulations when, compared to the fluxes themselves (Figure 6a,h,i).". This is more visible in figure 6 a,h,i and impacts what we see in figure 4. Likewise, the differences shown in figure S2 are all relatively small as shown in figure 6 and S3. Finally, the optimizations have little impact on the distribution of aboveground tree biomass in figure 5b when compared to the 32-tile run but yields an appreciable 23% decrease in computational load.** |
| Conclusion: this is a bit redundant with the previous section on implications. these two sections should just be combined for the conclusion. | **The conclusion and section 3.4 have been combined into a new section "4. Conclusion: Implications for representing disturbance and subgrid-scale heterogeneity in LSMs" to remove redundant text. (line 531 - 583)** |

---

## Author Response (AR2)

Dear Editor,

Please find our reply below. We copy your comments in full and our replies are in **bold blue**.

I thank the authors for their detailed response to the comments raised by the reviewers to their paper. The reviewers raised a few important scientific points that I think were addressed well, although I feel like some additional analysis is merited in one case. Due to this additional analysis, as well as a desire to confirm with reviewers that the revised text (combined) alleviates their concerns, I will ask one reviewer to look over the next version, and have thus requested "major revisions." However, I don't believe too much additional work is required.

One common substantive question raised by the reviewers has to do with whether the use of tiling (and choice of tiling setup) actually improves model accuracy relative to the real world. This is a very important consideration for many modeling papers, but it's less critical here, since this is a development & technical paper. As described on the GMD manuscript types webpage (https://www.geoscientific-model-development.net/about/manuscript_types.html#item2), such papers "usually include a significant amount of evaluation against standard benchmarks, observations, and/or other model output as appropriate." Here, the authors' model-model comparisons fall under the "other model output" category. They do also compare to observations, though, as illustrated in Fig. 5b.

I'm thus satisfied with the existing analyses given the authors' explanation. To clarify their goals, the authors have added and amended text in various places, removed the term "bias" in a few lines (see response to Reviewer 1, comment 4, last bullet point) and replaced "error" with "deviation" in, e.g., Fig. 4.

Another common and important point the reviewers mentioned is their concern about the abrupt transitions in disturbance regimes between the three periods in Fig. 2a-b, especially around 1985 for harvest (Fig. 2b). The authors explain in their response to Reviewer 1's comment 1 that (a) they chose to use distinct sets of derived records for these periods, as data were available; (b) that harvest is a relatively minor contribution to disturbance compared to fire; and (c) that all evaluated setups used the same disturbance history, so to some extent the possible worrying effects of abrupt regime transitions cancel out in model-model comparisons. Reviewer 1 also notes that the possible too-high harvest rate in the middle period means that the experimental results are conservative, if anything. I'm more or less convinced by the authors' arguments, but the fact that both reviewers raised the issue suggests that the authors could do more to convince readers.

To that end, I think it would be valuable to do some sort of additional experiment with at least a subset of the model setups (e.g., 1-tile/disturbed vs. 3 tiles vs. optimal) with an alternative disturbance scenario. Reviewer 2's suggestion of simply using the first observed value throughout the pre-observation period seems as good a choice as any. I suspect that we would

still see strong effects of tile count, which would bolster the conclusion that the tiling scheme is important.

**Thank you for your work with our article and follow-up comments. We appreciate your evaluation of our responses. We have provided a revised version of the manuscript including the additional analyses requested. A description of our further edits to the manuscript and relevant explanation follows.**

**We have revised the manuscript to include Figure S4 (copied below). This figure details a set of model runs using alternative pre-1985 fire and harvest scenarios. In these scenarios (figure S4a-b) pre-1985 fire and harvest are bias-corrected to match their respective means from the observed period. Based upon these model runs we can demonstrate that by 2017 the pre-1985 disturbance has a limited impact on the distribution of aboveground tree biomass in the model domain across our various simulation setups (figure S4c). We are also able to demonstrate that due to the construction of our model-on-model comparisons, the differences in the normalized response metric ($\Delta \bar{X}_{norm}$) between the runs using the alternative disturbance scenario ($\Delta \bar{X}_{norm,alternative}$; figure S4a-b) and the original disturbance scenario ($\Delta \bar{X}_{norm,original}$; Figure 2) are an order of magnitude lower than those shown in figure 6a/S3. The alternative disturbance scenario has a similar impact on all the simulations utilized in calculating $\Delta \bar{X}_{norm}$, which is then removed via the normalization (see eqn 12). These follow-up analyses demonstrate that our conclusions regarding the dynamic tiling parameters and the relative impacts of subgrid-scale heterogeneity are robust.**

**We have revised the main text to refer to these additional analyses and our conclusions therein. We have added a paragraph to the results near line 530 detailing the implications of this further analysis:**

> **"The relative impacts of subgrid-scale heterogeneity demonstrated herein are robust given our model-on-model approach, which cancels out pre-existing biases. This is evident in model runs using alternative pre-1985 fire and harvest scenarios that are bias-corrected to match their respective means from the observed period (Figure S4a-b). These alternative scenarios have a limited impact on the modeled statistical distributions of aboveground tree biomass (figure 5a, S4c). Moreover, because our statistical analysis focuses on the period during which disturbance observations are available (1985-2017) and because of the statistical metrics utilized in our model-on-model comparisons (eqn 12), the differences in $\Delta \bar{X}_{norm}$ with these alternative scenarios (Figure S4d) are an order of magnitude smaller than shown in figure 6a. Our evaluations provide insights into the impacts of subgrid-scale heterogeneity alone, further reinforcing its importance and the value of representing this heterogeneity within models."**

**We have also added a reference to figure S4 in the conclusion near line 562:**

> **"These results are strengthened by our model-on-model approach, which acts to cancel out pre-existing biases, to demonstrate the impacts of subgrid-scale heterogeneity, and discretization error alone (Figure S4)(Torres-Rojas et al., 2022;**

Curasi et al., 2022; Melton et al., 2017; Melton & Arora 2014; Moorcroft et al., 2001)."

**Figure:**

[Figure]

**Figure S4: Model runs using alternative historical harvest and fire scenarios. Plots of the alternative disturbance drivers over time. a) annual total burned and b) harvested area from 1740 - 2020. Observed indicates the period that uses the Landsat fire and harvest observations (Hermosilla et al., 2016, 2015a, b). Bias-corrected with constant values refers to the period where the inferred disturbance was bias-corrected using the mean of the Landsat observations. c) Weighted histogram of aboveground tree biomass for forested areas of Canada at the end of a selection of model runs including the 1-tile/not-disturbed run, 1-tile/disturbed, 3-tiles, and 32-tiles. The contributions of all forested subgrid areas weighted by their fractional area within the modeled region are considered. d) Plot of the difference in the normalized response metric between model runs using the alternative disturbance scenario ($\Delta\bar{X}_{norm,alternative}$) and the original disturbance scenario ($\Delta\bar{X}_{norm,original}$; Figure 2) for 1-tile/not-disturbed versus 1-tile/disturbed, 1-tile/disturbed versus 32-tile, for vegetation carbon (cVeg), soil carbon (cSoil), gross primary productivity (GPP), autotrophic respiration (Ra), heterotrophic respiration (Rh), leaf area index (LAI), sensible heat flux (HFSS), latent heat flux (HFLS), albedo (ALBS), fire emissions (fFire) and total deforested carbon (fDeforestTotal). All runs using >1 tile include disturbance. *denotes disturbance-related fluxes omitted in the 1-tile/not-disturbed model run.**